# Epistemic Uncertainty for Generated Image Detection

Jun Nie[1,2]    Yonggang Zhang[2,4]    Tongliang Liu[3]    Yiu-ming Cheung[2]
Bo Han[2]    Xinmei Tian[1]*

[1]MoE Key Laboratory of Brain-inspired Intelligent Perception and Cognition,
University of Science and Technology of China
[2]Hong Kong Baptist University    [3]Sydney AI Centre, The University of Sydney
[4]The Hong Kong University of Science and Technology

## Abstract

We introduce a novel framework for AI-generated image detection through epistemic uncertainty, aiming to address critical security concerns in the era of generative models. Our key insight stems from the observation that distributional discrepancies between training and testing data manifest distinctively in the epistemic uncertainty space of machine learning models. In this context, the distribution shift between natural and generated images leads to elevated epistemic uncertainty in models trained on natural images when evaluating generated ones. Hence, we exploit this phenomenon by using epistemic uncertainty as a proxy for detecting generated images. This converts the challenge of generated image detection into the problem of uncertainty estimation, underscoring the generalization performance of the model used for uncertainty estimation. Fortunately, advanced large-scale vision models pre-trained on extensive natural images have shown excellent generalization performance for various scenarios. Thus, we utilize these pre-trained models to estimate the epistemic uncertainty of images and flag those with high uncertainty as generated. Extensive experiments demonstrate the efficacy of our method. Code is available at https://github.com/tmlr-group/WePe.

## 1 Introduction

Recent advancements in generative models have revolutionized image generation, enabling the production of highly realistic images (Midjourney, 2022; Wukong, 2022; Rombach et al., 2022). Despite the remarkable capabilities of these models, they pose significant challenges, particularly the rise of deepfakes and manipulated content. The high degree of realism achievable by such technologies prompts urgent discussions about their potential misuse, especially in sensitive domains such as politics and economics. In response to these critical concerns, a variety of methodologies for detecting generated images have emerged. A prevalent strategy treats this task as a binary classification problem, necessitating the collection of extensive datasets comprising both natural and AI-generated images to train classifiers (Wang et al., 2020).

While existing methods have demonstrated notable successes, unseen generative models Wang et al. (2023) pose challenges for them in generalizing to images with distribution shifts. One promising avenue to enhance the robustness of detection capabilities involves constructing more extensive training sets by accumulating a diverse array of natural and generated images. However, these attempts are computationally intensive, requiring substantial datasets for effective classification. Besides, maintaining robust detection necessitates continually acquiring images generated by the latest generative models. And when the latest generative models are not open-sourced, acquiring a

---

*Correspondence to: Xinmei Tian (xinmei@ustc.edu.cn)

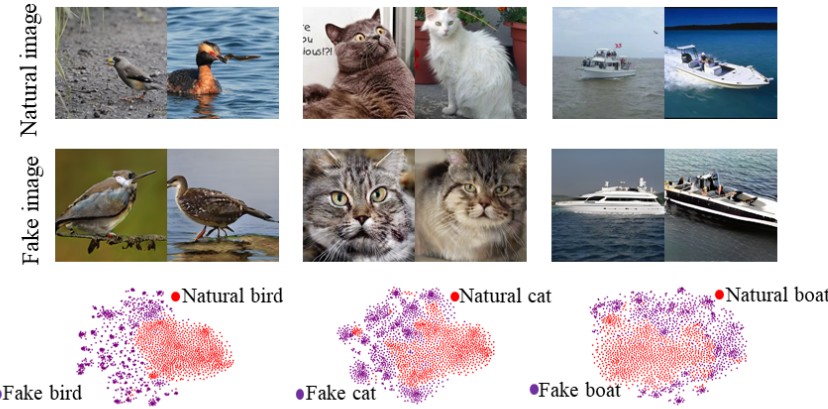

Figure 1: Models trained on a large number of natural images are capable of distinguishing between natural and generated images.

large number of generated images to train classifiers is challenging. This highlights the urgent need for a novel framework to detect generated images without reliance on generated images.

A recent work (Tan et al., 2024) shows that features extracted by ViT of CLIP (Radford et al., 2021) can be employed to separate natural and AI-generated images, motivating an effective approach to detecting images by training a binary classifier in the feature space of CLIP. This provides a promising direction to explore the possibility that large-scale foundational models already have the ability to capture the subtle differences between natural images and AI-generated images. As shown in Figure 1, even for images sampled from the same class, there are large distributional discrepancies in the feature space of DINOv2. This observation is consistent with an important metric for evaluating generative models-the FID score. FID score measures feature distribution discrepancy between natural and generated images on the Inception network (Szegedy et al., 2015). A FID score of 0 indicates that there is no difference between the two distributions. However, even on these simple networks such as Inception v3, advanced generative models like ADM still achieve an FID score of 11.84, not to mention that on powerful models such as DINOv2, we observe significant feature distribution discrepancy. However, despite these distributional differences, the inherent diversity of natural images precludes direct modeling of their distribution. This challenge motivates us to consider an alternative approach to reflect the distributional differences between natural and generated images.

In this study, we exploit the distributional disparity between natural and generated images by leveraging epistemic uncertainty to differentiate them. Epistemic uncertainty quantifies a model's confidence in its predictions, reflecting its knowledge of the data distribution. As the volume of training data increases, a model's epistemic uncertainty for in-distribution (ID) data diminishes. For a foundational model pre-trained on an extensive dataset of natural images, we posit that its epistemic uncertainty is lower for

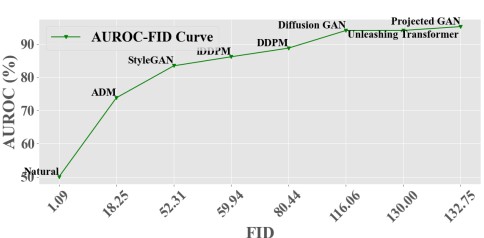

Figure 2: WePe reflects the distribution discrepancy between AI-generated and natural images.

natural images compared to generated ones, due to the model's alignment with the natural distribution. This perception is consistent with recent studies (Snoek et al., 2019; Schwaiger et al., 2020), which indicate that models tend to show increased uncertainty for out-of-distribution (OOD) samples. The challenge comes from efficiently obtaining the uncertainty of the model on the test samples. Classical approaches include Monte-Carlo Dropout (MC-Dropout) (Gal and Ghahramani, 2016) and Deep Ensembles (Lakshminarayanan et al., 2017). However, in our attempts, MC-Dropout obtains sub-optimal results (in Table 2), and it is challenging to train multiple large models independently for model ensemble. Instead, our theoretical results show that this uncertainty can be well captured by perturbing the weights of the models. As shown in Figure 3, when a moderate level of perturbation is applied, the natural image has consistent features on the model before and after the perturbation, but the generated image has large differences in features on the model before and after the perturbation.

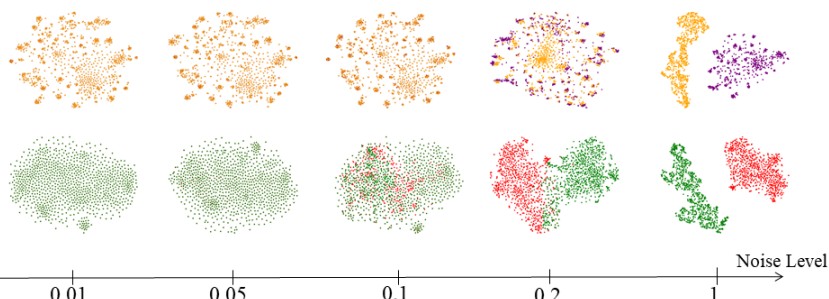

Figure 3: Natural and generated images exhibit distinct sensitivities to perturbations in model weights. A moderate perturbation (0.1) results in minimal changes to the features of the natural image, while the generated image shows significant differences.

In this paper, we propose a novel method for AI-generated image detection by weight perturbation (WePe). Our hypothesis is that the model has greater uncertainty in predicting the OOD sample compared to the ID sample, and that this uncertainty can be expressed through sensitivity to weight perturbations. For a large model trained on a large number of natural images, the natural images can be considered ID samples, while the generated images are considered OOD samples. Thus, the sensitivity of the samples to the weight perturbation of the large model can be an important indicator to determine whether the sample is generated by the generative models or not. As shown in Figure 2, we calculate FID scores between different types of generated images and natural images on DINOv2. It shows the performance of WePe strongly correlates with FID score, indicating the effectiveness of WePe in detecting distribution discrepancy.

We summarize our main contributions as follows:

- We provide a new perspective to detect AI-generated images by calculating predictive uncertainty. This is built on the fact that natural and generated images differ in data distribution, making it possible to employ uncertainty to represent the distribution discrepancy.

- Since the data distribution discrepancy between generated and natural images is reflected in feature distribution discrepancy on the vision model, we propose using a large vision model to compute prediction uncertainty to highlight this difference. The intuition is that large vision models are merely trained on natural images, making it possible to exhibit different uncertainties about natural and generated images. We capture this uncertainty through weight perturbation, enabling effective detection. (Eq. 13).

- Extensive experiments across multiple benchmarks show the proposed method surpasses existing methods, highlighting the efficacy of uncertainty in detecting AI-generated images.

## 2 Preliminaries

### 2.1 AI-generated image detection

The task of AI-generated image detection involves classifying a test image $\mathbf{x}$ as either a natural image or one produced by a generative model. Traditional supervised learning approaches rely on a curated dataset comprising both natural and AI-generated images to train a feature extractor and a binary classifier. This process is formalized as follows.

Let $\mathcal{D}^0 = \{\mathbf{x}_1^0, \mathbf{x}_2^0, \ldots, \mathbf{x}_{N^0}^0\}$ denote a set of $N^0$ AI-generated images, each labeled as $y = 0$ (generated), and $\mathcal{D}^1 = \{\mathbf{x}_1^1, \mathbf{x}_2^1, \ldots, \mathbf{x}_{N^1}^1\}$ denote a set of $N^1$ natural images, each labeled as $y = 1$ (natural). The objective is to learn a feature extractor $F(\cdot; \theta_F)$, parameterized by $\theta_F$, and a binary classifier $D(\cdot; \theta_D)$, parameterized by $\theta_D$, by minimizing a loss function $\ell(\cdot)$ over the training data:

$$F, D = \arg\min_{\theta_F, \theta_D} \ell\left(D\left(F(\mathbf{x}; \theta_F); \theta_D\right), y\right), \tag{1}$$

where $y \in \{0, 1\}$ represents the ground-truth label of the input image $\mathbf{x}$.

Upon completion of training, the feature extractor and classifier are used to compute a decision score for a test image $\mathbf{x}$, defined as $S(\mathbf{x}) = D(F(\mathbf{x}; \theta_F); \theta_D)$. The source of the image is determined by comparing this score to a predefined threshold $\tau$:

$$\mathrm{pred}(\mathbf{x}) = \begin{cases} \text{generated}, & \text{if } S(\mathbf{x}) < \tau, \\ \text{natural}, & \text{otherwise.} \end{cases} \tag{2}$$

The robustness and generalization of such methods depend critically on the size and diversity of the training dataset. To address this, techniques such as CNNspot employ sophisticated data augmentation strategies, including Gaussian blur and JPEG compression, to enhance the variability of the training data. In contrast, UnivFD adopts a different approach by utilizing the CLIP model as a fixed feature extractor and training only a single linear classification layer. Despite these innovations, both methods exhibit limited generalization when applied to images produced by unseen generative models.

## 2.2 Bayesian neural networks and uncertainty estimation

Traditional neural networks map inputs $\mathbf{x}$ to outputs $\mathbf{y}$ through a parameterized function $f(\mathbf{x}; \theta)$, where weights $\theta$ are optimized deterministically, offering no uncertainty quantification. Bayesian Neural Networks (Neal, 2012) address this by modeling $\theta$ as random variables with a prior distribution $p(\theta)$. Given a dataset $\mathcal{D}$, the posterior is computed via Bayes' theorem:

$$p(\theta|\mathcal{D}) = \frac{p(\mathcal{D}|\theta)p(\theta)}{p(\mathcal{D})}. \tag{3}$$

For a new input $\mathbf{x}^*$, the posterior predictive distribution:

$$p(\mathbf{y}^*|\mathbf{x}^*, \mathcal{D}) = \int \underbrace{p(\mathbf{y}^*|\mathbf{x}^*, \theta)}_{\text{Aleatoric}} \underbrace{p(\theta|\mathcal{D})}_{\text{Epistemic}} d\theta \tag{4}$$

captures prediction uncertainty.

Exact posterior inference is intractable due to the high dimensionality of $\theta$. Monte Carlo methods, such as MC Dropout (Gal and Ghahramani, 2016), approximate the posterior by using dropout during both training and testing and generating multiple stochastic predictions. Uncertainty is quantified through the variance or entropy of these predictions, reflecting two types:

- **Epistemic Uncertainty**: Uncertainty in model parameters, encoded in $p(\mathbf{w}|\mathcal{D})$, which decreases with more data.
- **Aleatoric Uncertainty**: Inherent noise in the data, modeled by the likelihood $p(\mathbf{y}|\mathbf{x}, \mathbf{w})$.

## 3 Uncertainty based AI-generated image detection

### 3.1 Motivation

Uncertainty estimation is a well-established field critical to deep learning practitioners, as it facilitates explicit handling of uncertain inputs and edge cases (Durasov et al., 2021; Everett et al., 2022). By quantifying model confidence, practitioners can make informed decisions, such as deferring high-uncertainty inputs to human evaluation in classification tasks, thereby improving model reliability and robustness. In this study, we do not propose a novel method for uncertainty estimation. Instead, we focus on distinguishing natural from generated images by leveraging epistemic uncertainty as a discriminative metric. Our approach is grounded in the well-established principle that (Lahlou et al., 2023; Gal and Ghahramani, 2016):

> *Epistemic uncertainty, which reflects a model's lack of knowledge, can be reduced by acquiring additional information.*

Formally, we adopt a Bayesian framework to quantify epistemic uncertainty through the posterior distribution over model parameters:

$$p(\theta|\mathcal{D}) = \frac{p(\mathcal{D}|\theta)p(\theta)}{p(\mathcal{D})}, \tag{5}$$

where $\mathcal{D} = \{(\mathbf{x}_i, \mathbf{y}_i)\}_{i=1}^N$ denotes the training dataset, with samples drawn i.i.d. from $p(\mathbf{y}|\mathbf{x}, \theta_0)$. Epistemic uncertainty is captured by the posterior variance:

$$\text{Var}(\theta|\mathcal{D}) = \mathbb{E}_{\theta \sim p(\theta|\mathcal{D})}[(\theta - \mathbb{E}[\theta|\mathcal{D}])^2]. \tag{6}$$

Under regularity conditions, the Bernstein-von Mises theorem (Van der Vaart, 2000) establishes that, as the sample size $N \to \infty$,

$$p(\theta|\mathcal{D}) \approx \mathcal{N}\left(\hat{\theta}_N, \frac{1}{N}I(\theta_0)^{-1}\right), \tag{7}$$

where $\hat{\theta}_N$ is the maximum likelihood estimate, and $I(\theta_0) = \mathbb{E}\left[\left(\frac{\partial \log p(\mathbf{y}|\mathbf{x}, \theta)}{\partial \theta}\right)^2 \big|_{\theta=\theta_0}\right]$ represents the Fisher information matrix. Here, $\theta_0$ approximates the true parameters of the natural image distribution, as learned by a pre-trained vision model. Consequently, $\text{Var}(\theta|\mathcal{D}) \propto \frac{1}{N}$, indicating that epistemic uncertainty decreases as the training data volume increases.

For a test input $\mathbf{x}^*$ drawn from a distribution $p_{\text{test}}(\mathbf{x}) \neq p_{\text{train}}(\mathbf{x})$, the predictive distribution is given by:

$$p(\mathbf{y}^*|\mathbf{x}^*, \mathcal{D}) = \int p(\mathbf{y}^*|\mathbf{x}^*, \theta)p(\theta|\mathcal{D}) \, d\theta. \tag{8}$$

This distribution exhibits elevated variance due to the misalignment between $p(\theta|\mathcal{D})$ and the parameters required to model $p_{\text{test}}(\mathbf{x})$. The predictive variance can be decomposed as:

$$\text{Var}(\mathbf{y}^*|\mathbf{x}^*, \mathcal{D}) = \mathbb{E}_{\theta \sim p(\theta|\mathcal{D})}[\text{Var}(\mathbf{y}^*|\mathbf{x}^*, \theta)] + \text{Var}_{\theta \sim p(\theta|\mathcal{D})}[\mathbb{E}(\mathbf{y}^*|\mathbf{x}^*, \theta)], \tag{9}$$

where the second term, $\text{Var}_{\theta \sim p(\theta|\mathcal{D})}[\mathbb{E}(\mathbf{y}^*|\mathbf{x}^*, \theta)]$, corresponds to epistemic uncertainty and dominates for out-of-distribution (OOD) samples. Under distribution shift, the Fisher information $I(\theta_0)$, defined with respect to $p_{\text{train}}(\mathbf{x})$, is misaligned with $p_{\text{test}}(\mathbf{x})$, limiting the reduction of posterior variance and resulting in persistently high epistemic uncertainty (Snoek et al., 2019).

Based on the above analysis, as well as the phenomenon of distributional shifts between natural and generated images on the foundational model (see Figures 1 and 8), we believe that the difference in the epistemic uncertainty of the foundational model on natural and generated images can be a valid indicator to distinguish between them.

## 3.2 Uncertainty estimation via weight perturbation

Classical methods of epistemic uncertainty estimation, such as Deep Ensembles and MC Dropout, can simply be viewed as using variance of multiple prediction results as an estimation of uncertainty $u(\mathbf{x})$:

$$\mu(\mathbf{x}) = \frac{1}{n}\sum_{t=1}^n \hat{y}_t(\mathbf{x}), u(\mathbf{x}) = \sigma^2 = \frac{1}{n}\sum_{t=1}^n (\hat{y}_t(\mathbf{x}) - \mu(\mathbf{x}))^2, \tag{10}$$

where, $\hat{y}_t$ denotes the $t$-th prediction. The multiple predictions of Deep Ensembles come from multiple independently trained neural networks, while the multiple predictions of MC Dropout come from the use of dropout during inference, which can be regarded as multiple prediction using neural networks with different structures.

In this study, we leverage DINOv2 for uncertainty estimation, capitalizing on two key advantages. First, DINOv2 is pre-trained on an extensive datasets, endowing it with robust generalization capabilities. This broad pre-training enables the model to effectively capture the distributional characteristics of natural images, thereby enhancing the generalizability of our proposed method across diverse scenarios. Second, as DINOv2 is exclusively pre-trained on natural images, it exhibits distinct uncertainty profiles when processing natural versus AI-generated images. This differential uncertainty provides a reliable basis for distinguishing between the two image types. DINOv2 is a self-supervised learning model, employing a student-teacher framework where the student model $\theta$ is trained to align with the representations of the teacher model $\theta_t$. Consequently, we utilize the feature similarity between the embeddings produced by the teacher and student models as the prediction:

$$\hat{y}(x) = f(\mathbf{x}; \theta)^\top f(\mathbf{x}; \theta_t), \tag{11}$$

where $f(\mathbf{x}; \theta)$ denotes the L2-normalized features of an input image $\mathbf{x}$ when inferring with the parameter $\theta$. However, estimating uncertainty with DINOv2 presents challenges in obtaining multiple

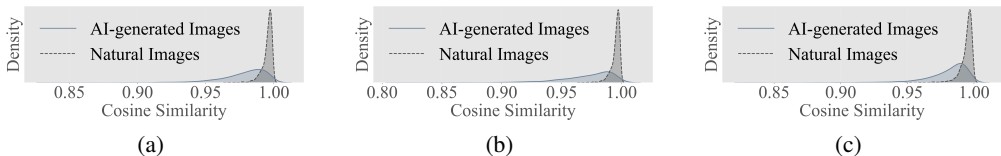

Figure 4: Comparison of cosine similarity between features on original and perturbed models. The generated images are from: (a) ADM, (b) BigGAN, and (c) DDPM.

predictions. First, as DINOv2 does not employ dropout during training, MC Dropout may yield suboptimal results (see Table 2). Second, training multiple DINOv2-scale models is computationally infeasible, rendering Deep Ensemble impractical. Instead, we use an alternative approach, weight perturbation, to obtain multiple predictions. Specifically, let $\theta_k$ denote the perturbed parameters of the student network, according to Eq. 10, the predictive uncertainty $u(\mathbf{x})$ can be calculated by,

$$u(\mathbf{x}) = \frac{1}{n} \sum_{k=1}^{n} [f(\mathbf{x};\theta_k)^\top f(\mathbf{x};\theta_t) - \sum_{j=1}^{n} \frac{f(\mathbf{x};\theta_j)^\top f(\mathbf{x};\theta_t)}{n}]^2, \qquad (12)$$

However, we cannot access the teacher model $\theta_t$, making it challenging to calculate the uncertainty. Moreover, even if it is available, introducing two models for calculation leads to low computation efficiency. Fortunately, we can calculate an upper bound of $u(\mathbf{x})$. This can be formalized by,

$$u(\mathbf{x}) \le \frac{1}{n} \sum_{k=1}^{n} \left\| f(\mathbf{x};\theta_k) - \frac{1}{n} \sum_{j} f(\mathbf{x};\theta_j) \right\|^2 \|f(\mathbf{x};\theta_t)\|^2 = 2 - \frac{2}{n} \sum_{k=1}^{n} f(\mathbf{x};\theta_k)^\top f(\mathbf{x};\theta), \quad (13)$$

where $\theta$ denotes the parameter of student model before injecting perturbation, and we leverage an unbiased assumption that the expectation $\mathbb{E}_{\theta_j} f(\mathbf{x};\theta_j)$ approaches the feature $f(\mathbf{x};\theta)$ extracted by the non-perturbed parameter $\theta$. Eq. 13 provides a simple approach to calculate the uncertainty without needing a teacher model used in the training phase of DINOv2. We provide an analysis of the validity of this upper bound in Appendix D. The insight of Eq. 13 is intuitive. Specifically, if an image $\mathbf{x}$ causes a high feature similarity between the original and perturbed parameter, the image leads to a low uncertainty and is more likely to be a natural image.

### 3.3 Theoretical analysis of the effectiveness of weight perturbation

In this section, we present a theoretical analysis elucidating why weight perturbations effectively distinguish between natural and AI-generated images. We establish a formal metric to quantify the sensitivity of neural network feature representations to parameter perturbations and demonstrate its differential behavior across natural images and AI-generated images.

To formalize this notion, we introduce the following definition of perturbation sensitivity:

**Definition 3.1** (Perturbation Sensitivity). For a neural network $f : \mathcal{X} \to \mathbb{R}^d$, parameterized by $\theta \in \mathbb{R}^p$, which maps an input $x \in \mathcal{X}$ to a feature vector $f(x;\theta)$, the sensitivity to parameter perturbations is defined as:

$$\text{sen}(x) = \|\nabla_\theta f(x;\theta)\|_F^2, \qquad (14)$$

where $\nabla_\theta f(x;\theta)$ denotes the Jacobian matrix of the feature mapping with respect to $\theta$, and $\|\cdot\|_F$ represents the Frobenius norm.

This metric captures the magnitude of variation in the feature representation induced by infinitesimal changes in the model parameters, providing a robust measure of sensitivity to weight perturbations.

We proceed to establish the differential sensitivity of the model across distributions through the following theorem:

**Theorem 3.2** (Differential Sensitivity). *Let a neural network $f(x;\theta)$ be trained on a large amount of natural images sampled from natural image distribution $\mathcal{D}^1$: $T = \{(x^1,y^1),(x^2,y^2),...,(x^n,y^n)\} \sim \mathcal{D}^1$. The expected sensitivity of the feature representations to parameter perturbations is lower for inputs drawn from $\mathcal{D}^1$ compared to those drawn from a generated image distribution $\mathcal{D}^0$, generated by generative models. Formally:*

$$\mathbb{E}_{x \sim \mathcal{D}^1} [sen(x)] \le \mathbb{E}_{x \sim \mathcal{D}^0} [sen(x)] . \qquad (15)$$

Table 1: AI-generated image detection performance on ImageNet. Values are percentages. **Bold** numbers are superior results. We compare training methods and training-free methods separately.

| Methods | ADM | | ADMG | | LDM | | DiT | | BigGAN | | GigaGAN | | StyleGAN XL | | RQ-Transformer | | Mask GIT | | Average | |
|---|---|---|---|---|---|---|---|---|---|---|---|---|---|---|---|---|---|---|---|---|
| | AUROC | AP | AUROC | AP | AUROC | AP | AUROC | AP | AUROC | AP | AUROC | AP | AUROC | AP | AUROC | AP | AUROC | AP | AUROC | AP |
| *Training Methods* | | | | | | | | | | | | | | | | | | | | |
| CNNspot | 62.25 | 63.13 | 63.28 | 62.27 | 63.16 | 64.81 | 62.85 | 61.16 | 85.71 | 84.93 | 74.85 | 71.45 | 68.41 | 68.67 | 61.83 | 62.91 | 60.98 | 61.69 | 67.04 | 66.78 |
| Ojha | 83.37 | 82.95 | 79.60 | 78.15 | 80.35 | 79.71 | 82.93 | 81.72 | 93.07 | 92.77 | 87.45 | 84.88 | 85.36 | 83.15 | 85.19 | 84.22 | 90.82 | 90.71 | 85.35 | 84.25 |
| DIRE | 51.82 | 50.29 | 53.14 | 52.96 | 52.83 | 51.84 | 54.67 | 55.10 | 51.62 | 50.83 | 50.70 | 50.27 | 50.95 | 51.36 | 55.95 | 54.83 | 52.58 | 52.10 | 52.70 | 52.18 |
| NPR | 85.68 | 80.86 | 84.34 | 79.79 | 91.98 | 86.96 | 86.15 | 81.26 | 89.73 | 84.46 | 82.21 | 78.20 | 84.13 | 78.73 | 80.21 | 73.21 | 89.61 | 84.15 | 86.00 | 80.84 |
| PatchCraft | 81.83 | 79.65 | 70.88 | 69.36 | 68.47 | 65.19 | 75.38 | 73.29 | 99.85 | 99.26 | 98.55 | 97.91 | 96.33 | 96.25 | 91.28 | 91.47 | 92.56 | 92.17 | 86.13 | 84.95 |
| FatFormer | 91.77 | 90.36 | 83.58 | 83.17 | **92.58** | **92.06** | 86.93 | 85.14 | 98.76 | 98.47 | 97.65 | 98.02 | 97.64 | 97.57 | 96.55 | 95.96 | 97.65 | 97.27 | 93.68 | 93.11 |
| DRCT | 90.26 | 90.07 | 85.74 | 83.85 | 90.24 | 89.88 | **88.27** | **89.06** | 95.87 | 94.99 | 86.89 | 86.12 | 89.11 | 88.39 | 92.38 | 92.41 | 94.44 | 94.47 | 90.36 | 89.92 |
| WePe* | **93.89** | **92.42** | **90.21** | **87.15** | 91.73 | 88.69 | 88.00 | 84.94 | **99.85** | **99.83** | **99.03** | **99.04** | **99.52** | **99.51** | **98.31** | **97.84** | **99.63** | **99.54** | **95.57** | **94.33** |
| *Training-free Methods* | | | | | | | | | | | | | | | | | | | | |
| AEROBLADE | 55.61 | 54.26 | 61.57 | 56.58 | 62.67 | 60.93 | 85.88 | 87.71 | 44.36 | 45.66 | 47.39 | 48.14 | 47.28 | 48.54 | 67.05 | 67.69 | 48.05 | 48.75 | 57.87 | 57.85 |
| RIGID | 87.16 | 85.08 | 80.09 | 77.07 | 72.43 | 69.30 | 70.40 | 65.94 | 90.08 | 89.26 | 86.39 | 84.11 | 86.32 | 85.44 | 90.06 | 88.74 | 89.30 | 89.25 | 83.58 | 81.58 |
| WePe | **89.79** | **87.32** | **83.20** | **78.80** | **78.47** | **73.50** | 77.13 | 71.21 | **94.24** | **93.64** | **92.15** | **90.29** | **93.86** | **92.86** | **93.50** | **91.47** | **89.55** | 86.25 | **87.99** | **85.04** |

(a)  (b)  (c)

Figure 5: Performance varies with perturbation intensity under different degradation mechanisms, including (a) JPEG compression, (b) Gaussian blur, and (c) Gaussian noise.

This theorem asserts that feature representations of natural images, optimized through training on $\mathcal{D}^1$, exhibit greater robustness to parameter perturbations than those of AI-generated images, which lie outside the training distribution. This differential sensitivity underpins the efficacy of weight perturbations in AI-generated image detection, enabling the model to distinguish natural images from AI-generated images. A rigorous proof of this theorem is provided in Appendix C.

### 3.4 Sharpening discriminative uncertainty through precise calibration

The original DINOv2 model reveals notable uncertainty disparities between natural and generated images. With access to the training dataset, fine-tuning can amplify this uncertainty gap, enhancing detection performance. Specifically, we introduce the following loss function:

$$\mathcal{L}(\theta) = \mathbb{E}_{x \in X^1} \left[ f(\mathbf{x};\theta)^\top f(\mathbf{x};\theta^*) \right] - \mathbb{E}_{x \in X^0} \left[ f(\mathbf{x};\theta)^\top f(\mathbf{x};\theta^*) \right], \tag{16}$$

where $\theta^* = \theta + P$ represents a perturbed parameter set, with $P$ being a randomized perturbation matrix drawn from a predefined distribution. This formulation builds on the uncertainty upper bound established in Eq. 13 to guide optimization. The loss encourages high confidence for natural images ($X^1$) and low confidence for generated images ($X^0$), thereby sharpening the model's ability to distinguish between the two classes. We call this method WePe*.

## 4 Experiments

### 4.1 Experiment setup

**Datasets and evaluation metrics.** Following previous works (He et al., 2024; Zhu et al., 2023), we evaluate the performance of WePe on ImageNet (Deng et al., 2009), LSUN-BEDROOM (Yu et al., 2015), GenImage (Zhu et al., 2023) and DRCT-2M (Chen et al., 2024), with the following evaluation metrics: (1) the average precision (AP), (2) the area under the receiver operating characteristic curve (AUROC) and (3) the classification accuracy (ACC).

**Baselines.** Following RIGID, we take both training-free methods and training methods as baselines. For training-free methods, we take RIGID (He et al., 2024) and AEROBLADE (Ricker et al., 2024) as baselines. For training methods, we take DIRE (Wang et al., 2023), CNNspot (Wang et al., 2020), Ojha (Ojha et al., 2023), PatchCraft (Zhong et al., 2023), FatFormer (Liu et al., 2024), DRCT (Chen et al., 2024) and NPR (Tan et al., 2024) as baselines. Besides, on GenImage, we also report the result of Frank (Frank et al., 2020), Durall (Durall et al., 2020), Patchfor (Chai et al., 2020), F3Net (Qian et al., 2020), SelfBlend (Shiohara and Yamasaki, 2022), GANDetection (Mandelli et al., 2022), LGrad (Tan et al., 2023), ResNet-50 (He et al., 2016), DeiT-S (Touvron et al., 2021), Swin-T (Liu et al., 2021), Spec (Zhang et al., 2019), GramNet (Liu et al., 2020).

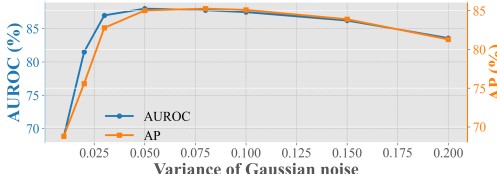

Figure 6: Performance varies with variance

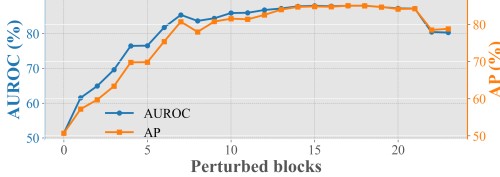

Figure 7: The effects of disturbed blocks.

**Experiment details.** To balance detection performance and efficiency, we use DINOv2 ViT-L/14. Due to the randomness of the added noise, we report the average results under five different random seeds and report the variance in Figure 10. In our experiments we find that perturbing the high layers may lead to a large corruption in the features of the natural images, resulting in sub-optimal

Table 2: The effect of perturbation type.

| Noise | AUROC | AP |
|---|---|---|
| Gaussian noise | 87.99 | 85.04 |
| Uniform noise | 89.06 | 86.32 |
| Laplace noise | 87.13 | 84.22 |
| MC Dropout | 81.63 | 79.71 |

results. Therefore, we do not perturb the high-level parameters. In DINOv2 ViT-L/14, the model has 24 transformer blocks, and we only perturb the parameters of the first 19 blocks with Gaussian perturbations of zero mean. The variance of Gaussian noise is proportional to the mean value of the parameters in each block, with the ratio set to 0.1.

## 4.2 Results

**Comparison with other baselines.** We conduct full comparative experiments on four benchmarks mentioned. As shown in Table 1, 10, 11 and 9, WePe achieves good detection performance on ImageNet, LSUN-BEDROOM, DRCT-2M and GenImage. Experimental results show the effectiveness of uncertainty estimation in detecting AI-generated images. It is worth noting that without any training, and relying only on the nature of the pre-trained model itself, WePe shows the potential to differentiate between natural and generated images. When further trained to amplify the uncertainty disparity between natural and generated images, WePe* achieves superior average performance compared to other methods and demonstrates robust generalization capabilities across diverse datasets. To further illustrate the effectiveness of our method, we count the difference in feature similarity between natural and generated images on the pre- and post-perturbation models. As shown in Figure 4, small perturbation of the model has less effect on the natural images than on generated images, resulting in higher feature similarity before and after the perturbation. The discrepancy effectively distinguishes the natural image from the generated image.

**Comparison under attacks.** In real-world scenarios, malicious actors may attempt to modify generated images to evade detection. To assess the robustness of the model under such conditions, we simulate an attack by introducing Gaussian noise (with a variance of $0.1$) to the generated images. Then the clean natural image and the attacked generated image are fed into the detector to determine whether the two can be reliably distinguished. Beyond spatial domain attacks, we also investigate attacks in the frequency domain, given the frequency differences between natural and generated images. As in Table 4, several detectors, such as NPR and RIGID, are vulnerable to this simple form of attack. In contrast, WePe, which identifies differences in the distributions of natural and generated images, demonstrates resilience to attacks, as the added noise further accentuates these distributional disparities, thereby enhancing WePe's capacity to differentiate between the two types of images.

## 4.3 Ablation study

In this section, we perform ablation experiments. Unless otherwise stated, experiments are conducted on ImageNet.

Table 3: The effect of models.

| model | AUROC | AP |
|---|---|---|
| DINOv2: ViT-S/14 | 72.83 | 71.63 |
| DINOv2: ViT-B/14 | 81.82 | 80.64 |
| DINOv2: ViT-L/14 | 87.99 | 85.04 |
| DINOv2: ViT-g/14 | 84.92 | 81.83 |
| CLIP: ViT-L/14 | 84.82 | 84.20 |

**Robustness to Image Perturbations** Robustness to various perturbations is a critical metric for detecting generated images. In real-world scenarios, images frequently undergo perturbations that can impact detection performance. Following RIGID, we assess the robustness of detectors against three types of perturbations: JPEG compression (with quality parameter $q$)), Gaussian blur (with standard deviation $\sigma$), and Gaussian noise (with standard deviation $\sigma$). As illustrated in Figure 5, training-free methods generally exhibit superior robustness compared to training-based methods, with our method achieving the best overall performance.

**Selecting which layers' parameters to perturb?** As shown in Figure 7, we explore the choice of which layers' parameters to perturb would achieve good performance. The horizontal coordinates in the graph indicate that the first $k$ blocks are perturbed, not the $kth$ block. The experimental results exhibit that our method obtains good performance when the parameters of the first 9 to the first 20 blocks are chosen to be perturbed. This demonstrates the robustness of our method. In practice, we can select the layers to be perturbed by a small set of natural and generated images. And when the generated images are not available, we can also use the probe to determine which layers are perturbed using only the natural image. We describe our method in Appendix O.

**The effect of perturbation type.** In experiments, model parameters are perturbed with Gaussian noise. We further explore other perturbation, such as adding uniform or Laplace noise to the weight. Besides, we also explore MC Dropout, i.e., using dropout during inference. As shown in Table 2, all three weight perturbation methods achieve good performance, and outperform MC Dropout.

**The impact of the degree of perturbation.** As shown in Figure 6, we explore the effect of the degree of perturbation on the performance of WePe. It can be seen that WePe is quite robust to the level of perturbation noise. It is only when the noise is very large or very small that it leads to a degradation in performance. When the noise level is small, the features obtained before and after the model perturbation are extremely similar, while when the noise level is very large, the features obtained before and after the model perturbation are extremely dissimilar, and these two cases will result in the inability to effectively differentiate between natural and generated images.

**The effect of models.** In our experiments, we mainly used DINOv2 ViT-L/14 to extract features. We further explore the effect of using other models of DINOv2, including ViT-S/14, ViT-B/14, and ViT-g/14. In addition to this, we conduct experiments on the CLIP:ViT-L/14. As shown in Table 3, the performance on CLIP is not as good as on DINOv2. We hypothesize that the difference comes from the training approach of these models. CLIP learns features using image captions as supervision, which may make the features more focused on semantic information, whereas DINOv2 learns features only from images, which makes it more focused on the images themselves, and thus better able to capture subtle differences in natural and generated images.

# 5 Related work

**AI-Generated images detection.** Recent advancements in generative models, such as those by (Brock et al., 2019; Ho et al., 2020), have led to the creation of highly realistic images, highlighting the urgent need for effective algorithms to distinguish between natural and generated images. Prior research, including works by (Frank et al., 2020; Marra et al., 2018), primarily focuses on developing specialized binary classification neural networks to differentiate between natural and generated images. Notably, CNNspot (Wang et al., 2020) demonstrates that a standard image classifier trained on ProGAN can generalize across various architectures when combined with specific data augmentation techniques. NPR (Tan et al., 2024) introduces the concept of neighboring pixel relationships to capture differences between natural and generated images. PatchCraft (Zhong et al., 2023) proposes an efficient AI-generated image detector by exploring texture patch artifacts. FatFormer (Liu et al., 2024) introduces a forgery-aware adaptive Transformer that adapts a pre-trained CLIP model to effectively discern and integrate local forgery traces from both image and frequency domains. DRCT (Chen et al., 2024) presents a Diffusion Reconstruction Contrastive Training framework to improve the generalizability of synthetic image detectors by training them to distinguish real images from their high-quality diffusion-based reconstructions. Although these methods show superior performance on generators in the training set, they often do not generalize well to unknown generators. In addition to this, training-based methods are susceptible to small perturbations in the image. For this reason, recently, some training-free methods have been proposed. AEROBLADE (Ricker et al., 2024) calculates the reconstruction error with the help of the autoencoder used in latent diffusion models (Rombach et al., 2022). RIGID (He et al., 2024) finds that natural images are more robust to small noise perturbations than generated images in the representation space of the vision foundation models and exploits this property for detection. However, these methods usually make overly strong assumptions about natural or generated images, leading to insufficient generalization. In our paper, we propose a training-free detection method through uncertainty analysis. Based on the widespread phenomenon that generated images have greater uncertainty than natural images on models trained with natural images, our method achieves robust detection performance.

**Uncertainty estimation.** Uncertainty estimation in machine learning has seen significant advancements in recent years. (Gal and Ghahramani, 2016) introduces Monte Carlo Dropout (MC Dropout), which uses dropout at inference to estimate uncertainty from the variance of multiple predictions. (Lakshminarayanan et al., 2017) develops deep ensembles, demonstrating improved uncertainty estimates through training multiple model independently with different initializations. Recent work by (Snoek et al., 2019) analyzes the calibration of uncertainty in deep learning models, highlighting the importance of reliable uncertainty measures. Additionally, (Guo et al., 2017) explore the use of temperature scaling to enhance the calibration of model predictions. (Blundell et al., 2015) introduce Bayes by Backprop, a method for estimating weight uncertainty in neural networks by modeling the posterior distribution over weights using variational inference, improving model generalization and robustness. Similarly, (Ferrante et al., 2024) leverage weight perturbation techniques to estimate neural network uncertainty, demonstrating improved classification accuracy through robust uncertainty quantification. (Pearce et al., 2020) explore distribution-free uncertainty estimation, using conformal prediction and quantile regression to estimate bounds on aleatoric uncertainty. (Chan et al., 2024) introduce hyper-diffusion models, allowing to accurately estimate both epistemic and aleatoric uncertainty with a single model.

## 6    Conclusion

In this work, to effectively address the challenges of detecting AI-generated images, we propose a novel approach that leverages predictive uncertainty as a key metric. Our findings reveal that by analyzing the discrepancies in distribution between natural and AI-generated images, we can significantly enhance detection performance. The use of large-scale pre-trained models allows for accurate computation of predictive uncertainty, enabling us to identify images with high uncertainty as likely AI-generated. Our method achieves robust detection performance in a simple untrained manner. Overall, our approach demonstrates a promising direction for improving AI-generated image detection and mitigating potential risks associated with their misuse. Future work could delve deeper into refining the predictive models and exploring additional features that could further enhance detection accuracy.

## Acknowledgments

This work was supported in part by NSFC No. 62222117. JN and BH were supported by NSFC General Program No. 62376235, RGC Young Collaborative Research Grant No. C2005-24Y, RGC General Research Fund No. 12200725, Guangdong Basic and Applied Basic Research Foundation Nos. 2022A1515011652 and 2024A151501239, and HKBU CSD Departmental Incentive Scheme. TLL was partially supported by the following Australian Research Council projects: FT220100318, DP220102121,LP220100527,LP220200949. YGZ was funded by Inno HK Generative AI R&D Center. YCM was supported by the RGC Senior Research Fellow Scheme under the grant: SRFS2324-2S02.

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

# A   Limitation

A key limitation of the proposed method lies in its reliance on the assumption of distinct uncertainty profiles between natural and generated images. As generative models continue to advance, the uncertainty gap between these image types, as captured by the foundational model, may progressively narrow. This convergence could undermine the method's effectiveness, necessitating additional calibration techniques such as the proposed WePe$^*$, to amplify the differentiation in uncertainty.

# B   Social impacts

The proposed AI-generated image detection method contributes to mitigating societal risks posed by generative model fraud. By improving the ability to identify synthetic media, such as deepfakes, this work helps combat disinformation and enhances trust in digital content, particularly in critical domains like journalism and legal evidence.

# C   Proof of THEOREM 3.2

*Proof.* Consider a neural network $f(x; \theta) : \mathcal{X} \to \mathbb{R}^d$, parameterized by $\theta \in \mathbb{R}^p$, which maps an input image $x$ to a feature vector $f(x; \theta)$. The model is trained on the natural image distribution $\mathcal{D}^1$ by minimizing a loss function:

$$\mathcal{L}(\theta) = \mathbb{E}_{x \sim \mathcal{D}^1} \left[ \ell(f(x; \theta), y) \right], \tag{17}$$

where $\ell$ is the loss, and $y$ is the label. The optimal parameters are denoted $\theta^*$.

To capture sensitivity, we define a loss function that measures the change in the feature vector under parameter perturbations:

$$\ell(x, \theta, \xi) = \|f(x; \theta + \xi) - f(x; \theta)\|_2^2, \tag{18}$$

where $\xi \sim \mathcal{N}(0, \sigma^2 I)$ represents a Gaussian perturbation with variance $\sigma^2$. For small $\xi$, we approximate:

$$f(x; \theta + \xi) \approx f(x; \theta) + \nabla_\theta f(x; \theta)^\top \xi, \tag{19}$$

so:

$$\ell(x, \theta, \xi) \approx \|\nabla_\theta f(x; \theta)^\top \xi\|_2^2 = \xi^\top J(x)^\top J(x) \xi, \tag{20}$$

where $J(x) = \nabla_\theta f(x; \theta)$ is the Jacobian matrix. The expected loss over perturbations is:

$$\mathbb{E}_\xi[\ell(x, \theta, \xi)] \approx \mathbb{E}_\xi \left[ \xi^\top J(x)^\top J(x) \xi \right] = \sigma^2 \mathrm{tr} \left( J(x)^\top J(x) \right) = \sigma^2 \|J(x)\|_F^2. \tag{21}$$

Thus, the expected sensitivity is proportional to the expected loss:

$$\mathbb{E}_{x \sim \mathcal{D}} \left[ \mathrm{sen}(x) \right] = \mathbb{E}_{x \sim \mathcal{D}} \left[ \|J(x)\|_F^2 \right] = \frac{1}{\sigma^2} \mathbb{E}_{x \sim \mathcal{D}} \mathbb{E}_\xi[\ell(x, \theta, \xi)] \geq 0. \tag{22}$$

According to PAC-Bayes theory (McAllester, 1998), we consider a prior distribution $P$ over parameters $\theta$, typically $P = \mathcal{N}(\theta_0, \sigma_0^2 I)$, and a posterior distribution $Q$, typically $Q = \mathcal{N}(\theta^*, \sigma^2 I)$, where $\theta^*$ is the trained parameter. The PAC-Bayes theorem provides a bound on the expected loss under $Q$.

For a loss function $\ell$, the standard PAC-Bayes bound states that, with probability at least $1 - \delta$ over the draw of a training set $T = \{(x_i, y_i)\}_{i=1}^N \sim \mathcal{D}^N$, for any posterior $Q$:

$$\mathbb{E}_{\theta \sim Q} \mathbb{E}_{(x,y) \sim \mathcal{D}}[\ell(\theta, x, y)] \leq \mathbb{E}_{\theta \sim Q} \left[ \frac{1}{N} \sum_{i=1}^N \ell(\theta, x_i, y_i) \right] + \sqrt{\frac{\mathrm{KL}(Q\|P) + \ln \frac{N}{\delta}}{2(N-1)}}, \tag{23}$$

where $\mathrm{KL}(Q\|P)$ is the Kullback-Leibler divergence between $Q$ and $P$.

Here, we adapt the loss to $\ell(x, \theta, \xi) = \|f(x; \theta + \xi) - f(x; \theta)\|_2^2$. Since $\xi$ is a perturbation, we consider the expected loss over $\xi$, which can be formed as:

$$\ell'(x, \theta) = \mathbb{E}_{\xi \sim \mathcal{N}(0, \sigma^2 I)} \left[ \|f(x; \theta + \xi) - f(x; \theta)\|_2^2 \right] \approx \sigma^2 \|J(x)\|_F^2. \tag{24}$$

Since $\ell'(x, \theta)$ depends on $\theta$, we approximate by evaluating at $\theta^*$, and consider the empirical sensitivity on the training set $T \sim \mathcal{D}^1$:

$$\hat{\ell}'(T, \theta) = \frac{1}{N} \sum_{i=1}^{N} \mathbb{E}_\xi \left[ \|f(x_i; \theta + \xi) - f(x_i; \theta)\|_2^2 \right]. \tag{25}$$

According to Eq. (23), we have the following inequality:

$$\mathbb{E}_{\theta \sim Q} \mathbb{E}_{x \sim \mathcal{D}}[\ell'(x, \theta)] \leq \mathbb{E}_{\theta \sim Q}[\hat{\ell}'(T, \theta)] + \sqrt{\frac{\mathrm{KL}(Q\|P) + \ln \frac{N}{\delta}}{2(N-1)}}. \tag{26}$$

For $\mathcal{D} = \mathcal{D}^1$, the training set $T \sim \mathcal{D}^1$, and the model is optimized at $\theta^*$. The empirical sensitivity is:

$$\hat{\ell}'(T, \theta^*) \approx \frac{\sigma^2}{N} \sum_{i=1}^{N} \|J(x_i)\|_F^2.$$

Since the model is optimized on $T \sim \mathcal{D}^1$, the training process minimizes the loss landscape's curvature around $\theta^*$, leading to a flat minimum (Hochreiter and Schmidhuber, 1997), which means the empirical sensitivity tends to 0. At the same time, as $n$ tends to infinity, resulting in the expectation sensitivity converging to the empirical sensitivity. Therefore, we have:

$$\mathbb{E}_{x \sim \mathcal{D}^1} \left[ \sigma^2 \|J(x)\|_F^2 \right] \approx 0, \tag{27}$$

which is consistent with our experimental results observed in Figure 3.

For $\mathcal{D} = \mathcal{D}^0$, we evaluate the expected sensitivity on generated images, but the training set remains $T \sim \mathcal{D}^1$. Since $\mathcal{D}^0$ is not optimized, the feature representations for $x \sim \mathcal{D}^0$ lie in regions of higher curvature in the loss landscape, leading to larger singular values of $J(x)$. According to Eq. (22) and Eq. (27), we have:

$$\mathbb{E}_{x \sim \mathcal{D}^0} \left[ \sigma^2 \|J(x)\|_F^2 \right] | \geq 0 \approx \mathbb{E}_{x \sim \mathcal{D}^1} \left[ \sigma^2 \|J(x)\|_F^2 \right], \tag{28}$$

which implies $\mathbb{E}_{x \sim \mathcal{D}^1} \left[ \mathrm{sen}(x) \right] \leq \mathbb{E}_{x \sim \mathcal{D}^0} \left[ \mathrm{sen}(x) \right]$. Thus we complete this proof. $\qquad \square$

## D   Analysis of the Upper Bounds for Uncertainty Estimation

In Eq. 13, we employ an upper bound to estimate uncertainty. Next, we provide an analysis to establish the validity of this estimation. The inequality is obtained through the Cauchy-Schwarz inequality, and we can explore the validity of the upper bound by calculating the difference between the two sides of the inequality. We define the difference:

$$\triangle = \|f(\mathbf{x}; \theta_k) - \frac{1}{n} \sum_{j=1}^{n} f(\mathbf{x}; \theta_j)\|^2 \cdot \|f(\mathbf{x}; \theta_t)\|^2 - \|(f(\mathbf{x}; \theta_k) - \frac{1}{n} \sum_{j=1}^{n} f(\mathbf{x}; \theta_j)) \cdot (f(\mathbf{x}; \theta_t))\|^2 \tag{29}$$

Thus, the difference $\triangle$ can be written as:

$$\triangle = \left\| f(\mathbf{x}; \theta_k) - \frac{1}{n} \sum_{j=1}^{n} f(\mathbf{x}; \theta_j) \right\|^2 \cdot \|f(\mathbf{x}; \theta_t)\|^2 - \left( \left\| f(\mathbf{x}; \theta_k) - \frac{1}{n} \sum_{j=1}^{n} f(\mathbf{x}; \theta_j) \right\| \|f(\mathbf{x}; \theta_t)\| \cos \theta \right)^2 \tag{30}$$

where $\theta$ is the angle between $f(\mathbf{x}; \theta_t)$ and $f(\mathbf{x}; \theta_k) - \frac{1}{n} \sum_j^n f(\mathbf{x}; \theta_j)$.

And finally, we obtain the following expression:

Table 4: Performance on ImageNet under frequency domain attacks (FDA) and spatial domain attacks (SDA). We report AUROC.

| Methods | ADM FDA | ADM SDA | ADMG FDA | ADMG SDA | LDM FDA | LDM SDA | DiT FDA | DiT SDA | BigGAN FDA | BigGAN SDA | GigaGAN FDA | GigaGAN SDA | StyleGAN XL FDA | StyleGAN XL SDA | RQ-Transformer FDA | RQ-Transformer SDA | Mask GIT FDA | Mask GIT SDA | Average FDA | Average SDA | Δ FDA | Δ SDA |
|---|---|---|---|---|---|---|---|---|---|---|---|---|---|---|---|---|---|---|---|---|---|---|
| *Training Methods* | | | | | | | | | | | | | | | | | | | | | | |
| CNNspot | 59.47 | 52.15 | 59.88 | 51.86 | 58.41 | 51.05 | 55.43 | 51.29 | 71.85 | 53.87 | 58.54 | 50.36 | 60.27 | 51.52 | 54.74 | 48.71 | 59.45 | 50.32 | 59.78 | 51.24 | -7.26 | -15.80 |
| Ojha | 85.16 | 85.28 | 84.09 | 83.67 | 85.94 | 84.97 | 85.14 | 84.33 | 92.17 | 94.77 | 85.39 | 90.18 | 88.16 | 86.73 | 85.22 | 86.38 | 88.71 | 91.75 | 86.17 | 87.56 | +0.82 | +2.21 |
| DIRE | 53.82 | 52.93 | 51.19 | 54.89 | 53.18 | 50.33 | 54.17 | 49.37 | 52.59 | 50.02 | 55.54 | 51.95 | 52.44 | 50.44 | 54.62 | 48.15 | 52.11 | 51.29 | 53.30 | 51.04 | +0.60 | -1.66 |
| NPR | 83.01 | 46.99 | 82.03 | 46.59 | 86.64 | 46.22 | 81.22 | 45.22 | 84.41 | 46.31 | 81.53 | 46.51 | 82.86 | 45.78 | 82.49 | 47.65 | 86.18 | 45.42 | 83.37 | 46.30 | -2.63 | -39.70 |
| WePe* | 96.42 | 93.86 | 93.67 | 88.58 | 94.73 | 84.90 | 91.47 | 81.93 | 99.64 | 96.67 | 98.54 | 94.69 | 99.46 | 97.33 | 98.49 | 95.64 | 98.67 | 91.79 | 96.79 | 91.71 | +1.22 | -3.86 |
| *Training-free Methods* | | | | | | | | | | | | | | | | | | | | | | |
| AEROBLADA | 40.30 | 23.08 | 42.87 | 27.87 | 45.51 | 28.18 | 43.93 | 28.77 | 41.84 | 27.09 | 42.70 | 29.05 | 47.01 | 32.78 | 51.62 | 31.96 | 51.62 | 32.71 | 44.90 | 29.05 | -12.97 | -28.82 |
| RIGID | 80.79 | 35.45 | 73.01 | 34.37 | 65.66 | 33.83 | 63.09 | 33.37 | 77.99 | 35.60 | 73.91 | 35.32 | 73.23 | 34.58 | 83.01 | 36.19 | 76.62 | 34.36 | 74.15 | 34.77 | -9.43 | -48.81 |
| WePe | 91.22 | 93.78 | 85.70 | 88.73 | 80.87 | 83.78 | 79.02 | 80.19 | 93.32 | 93.31 | 91.69 | 92.46 | 93.96 | 94.42 | 93.95 | 95.11 | 88.44 | 88.84 | 88.68 | 90.07 | +0.69 | +2.08 |

$$\triangle = \left\| f\left(\mathbf{x}; \theta_k\right) - \frac{1}{n} \sum_{j=1}^{n} f\left(\mathbf{x}; \theta_j\right) \right\|^2 \cdot \left\| f\left(\mathbf{x}; \theta_t\right) \right\|^2 \left(1 - \cos^2 \theta\right) \tag{31}$$

$$= \left\| f\left(\mathbf{x}; \theta_k\right) - \frac{1}{n} \sum_{j=1}^{n} f\left(\mathbf{x}; \theta_j\right) \right\|^2 \cdot \left\| f\left(\mathbf{x}; \theta_t\right) \right\|^2 \sin^2 \theta \tag{32}$$

This result indicates that the smaller the angle between $f\left(\mathbf{x}; \theta_k\right) - \frac{1}{n} \sum_{j}^{n} f\left(\mathbf{x}; \theta_j\right)$ and $f\left(\mathbf{x}; \theta_t\right)$, the tighter the upper bound. In our experiment, the perturbation applied to the model was extremely small, causing the image features to remain virtually unchanged, resulting in an extremely weak deviation between $f\left(\mathbf{x}; \theta_k\right) - \frac{1}{n} \sum_{j}^{n} f\left(\mathbf{x}; \theta_j\right)$ and $f\left(\mathbf{x}; \theta_t\right)$, thereby enabling the upper bound to provide a reliable basis for ranking the instances.

## E Performance under attacks.

As presented in Table 4, we evaluate the robustness of the proposed method against malicious manipulations of generated images. Specifically, for the FDA and SDA scenarios, we apply zero-mean Gaussian noise with a standard deviation of 0.1 to perturb the generated images. The results demonstrate that our method exhibits strong robustness to these perturbations.

## F Discussion on distribution discrepancy

In this paper, the core assumption we make is that there is data distribution discrepancy between natural and generated images. This assumption is valid for current generative models and has been confirmed by many works (Tan et al., 2024; Ricker et al., 2024). This assumption is also the foundation of many generative image detection methods (we cannot distinguish between images that are indistinguishable).

Secondly, we observe that this discrepancy in data distribution can be captured by the representation space of a vision model pre-trained on a large number of natural images, i.e., there is feature distribution discrepancy between the generated and natural images, as shown in Figure 8. However, this remains an observation, and we have not found theoretical proof despite reviewing the literature. We only observe a similar phenomenon in UnivFD (Ojha et al., 2023), where the feature distribution discrepancy is observed in the representation space of CLIP:ViT-L/14.

That said, we can confirm the existence of feature distribution discrepancy of generated and natural images based on an important metric for evaluating generative models, the FID score. The FID score measures the feature distribution discrepancy between natural and generated images on the Inception network (Szegedy et al., 2015). When the FID score is 0, it indicates that the two distributions do not differ. However, even on these simple networks such as Inception v3, advanced generative models like ADM still achieve an FID score of 11.84, not to mention that on powerful models such as DINOv2, we observe significant feature distribution discrepancy.

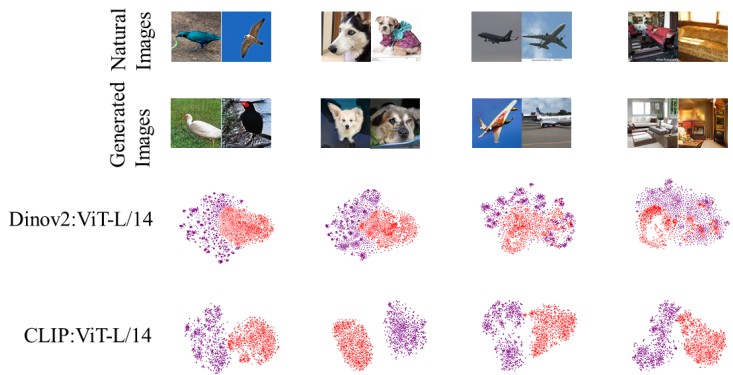

Figure 8: Feature distribution discrepancy between the generated and natural images on DINOv2 and CLIP. • and • represent the feature of natural images and AI-generated images on the corresponding models.

Table 5: Measuring feature distribution discrepancy with FID scores.

| Models | Natural | ADM | StyleGAN | iDDPM | DDPM | Diffusion GAN | Unleashing Transformer | Projected GAN |
|---|---|---|---|---|---|---|---|---|
| FID score | 1.09 | 18.25 | 52.31 | 59.94 | 80.44 | 116.06 | 130.00 | 132.75 |
| AUROC | 50.00 | 73.85 | 83.50 | 86.23 | 88.84 | 94.16 | 94.18 | 95.34 |

## G  Measuring feature distribution discrepancy with FID scores

We further use the "FID" score to measure the difference in feature distribution between natural and generated images. To avoid the effects of categories, we compute the FID scores using the DINOv2 model on the LSUN-BEDROOM benchmark. For each category of images, we randomly select 5000 images for calculation. In addition to calculating the FID scores between natural images and generated images, we also calculate the FID scores between natural images and natural images. As shown in Table 5, the FID scores between natural images and generated images are significantly higher than the FID scores between natural images and natural images. Moreover, there is a clear positive correlation between the detection performance of WePe and the FID score. This result fully explains the existence of feature distribution discrepancy between natural and generated images on DINOv2, and demonstrates that WePe can effectively detect the feature distribution discrepancy.

## H  WePe on large multi-modal models

In addition to CLIP, we further test the performance of WePe on BLIP (Li et al., 2022). As shown in Table 7, the performance of WePe is unsatisfactory on these multimodal models, which may be due to the fact that the image features of the multimodal models are more focused on semantic information, in line with our discussions.

Table 6: Comparison of detection times.

| Method | Time (s) |
|---|---|
| AEROBLADE | 17.6 |
| RIGID | 3.7 |
| WePe | 4.5 |

Table 7: WePe on large multi-modal models.

| Model | AUROC | AP |
|---|---|---|
| DINOv2 | 87.99 | 85.04 |
| BLIP | 68.25 | 64.68 |

## I  Comparison of computational costs.

Our method use a perturbed pre-trained model that is fixed during inferring all test samples. Thus, our method can be processed within two forward passes. This is equal to the cost of RIGID that requires two forward passes for clean and noisy images. However, RIGID can concatenate clean and noisy images in a mini batch and obtain detection results by with a single forward pass. AEROBLADE requires only one forward pass, but it needs to compute the reconstruction error of the image. This

takes a long time to reconstruct at the pixel level. Besides, AEROBLADE needs to use a neural network to compute the LPIPS score, leading to computational complexity. As shown in Table 6, we compare the time required to detect 100 images under the same conditions. Since AEROBLADE needs to calculate the image reconstruction error, it has the lowest detection efficiency. RIGID can obtain detection results in a single forward pass by concatenating clean and noisy images, whereas WePe requires two forward passes, which results in WePe's detection efficiency being inferior to RIGID's. However, WePe can be parallelized across two devices to obtain the detection results in a single forward pass. In practice, WePe has high computational efficiency and is suitable for large-scale applications.

## J  WePe with multiple perturbation

In our experiments, taking into account the detection efficiency, we perturb the model only once, and then calculate the feature similarity of the test samples on the clean and perturbed models. We further experiment with multiple perturbations and use the mean of the feature similarity of the test samples on the clean model and all the perturbed models as the criterion for determining whether or not the image is generated by the generative models. As shown in Figure 10, multiple perturbations can further improve performance.

## K  Comparison with OOD detection and uncertainty quantification methods

- Contrast with OOD Detection: Conventional OOD detection (Sun and Li, 2022; Djurisic et al., 2023; Nie et al., 2024) usually relies on Maximum Softmax Probability (MSP) scores, exploiting fixed ID categories to identify OOD samples with low probabilities. In contrast, AI-generated detection involves diverse, unbounded categories, rendering MSP scores ineffective. WePe introduces a novel uncertainty estimation approach tailored for detecting generated images, achieving robust performance where traditional OOD methods falter. As shown in Table 8, we further use the ImageNet pre-trained classification model and used MSP and entropy as the scoring function to evaluate their performance on the AI-generated image detection task. The results show that these methods fail.

- Contrast with Uncertainty Quantification: Standard techniques like MC-Dropout and Deep-Ensemble are ill-suited for DINOv2. Training multiple DINOv2 models for DeepEnsemble is computationally infeasible, and the absence of dropout in DINOv2 undermines MC-Dropout's efficacy. Our proposed weight perturbation method overcomes these limitations, delivering a practical and effective uncertainty estimation tailored to DINOv2's architecture, as validated by our experiments.

## L  Analysis of Failure Cases

WePe leverages the differential epistemic uncertainty exhibited by pre-trained models, such as DINOv2, when processing natural versus generated images to differentiate between them. Specifically, DINOv2, having been trained on an extensive dataset of natural images, demonstrates lower epistemic uncertainty for such images. Images that diverge from the distribution of the training dataset—despite not being generated by a gen-

Table 8: The results of using classification models.

| model | score | AUROC | AP |
|---|---|---|---|
| ResNet18 | MSP | 48.85 | 49.23 |
| ResNet18 | entropy | 51.76 | 50.28 |
| ViT-L/16 | MSP | 63.23 | 60.16 |
| ViT-L/16 | entropy | 65.79 | 61.97 |

erative model—tend to elicit higher uncertainty from DINOv2, leading to their misclassification as generated images. As shown in Figure 9, we visualized the feature shifts in DINOv2's representations for a set of cartoon images, which were not produced by generative models, before and after weight perturbation. These images, due to their deviation from the training natural distribution, exhibited significant feature shifts post-perturbation, rendering them indistinguishable from generated images in our analysis.

## M  Experiment results on GenImage, LSUN-BEDROOM and DRCT-2M

As shown in Table 9, Table 10 and Table 11, our method achieves good performance on GenImage LSUN-BEDROOM and DRCT-2M, confirming the robustness of the proposed method.

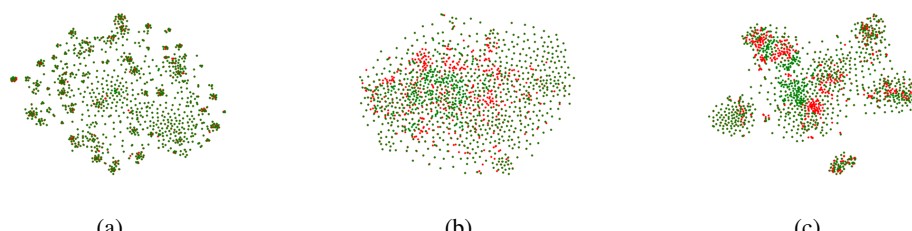

|     (a)     |     (b)     |     (c)     |

Figure 9: Feature shifts after model perturbation. Images are sampled from the following distributions: (a) natural image distribution, (b) AI-generated image distribution, and (c) cartoon image distribution.

Table 9: AI-generated image detection performance (ACC, %) on GenImage.

| Methods | Midjourney | SD V1.4 | SD V1.5 | ADM | GLIDE | Wukong | VQDM | BigGAN | Average |
|---|---|---|---|---|---|---|---|---|---|
| | | | | Training Methods | | | | | |
| ResNet-50 | 54.9 | 99.9 | 99.7 | 53.5 | 61.9 | 98.2 | 56.6 | 52.0 | 72.1 |
| DeiT-S | 55.6 | 99.9 | 99.8 | 49.8 | 58.1 | 98.9 | 56.9 | 53.5 | 71.6 |
| Swin-T | 62.1 | 99.9 | 99.8 | 49.8 | 67.6 | 99.1 | 62.3 | 57.6 | 74.8 |
| CNNspot | 52.8 | 96.3 | 95.9 | 50.1 | 39.8 | 78.6 | 53.4 | 46.8 | 64.2 |
| Spec | 52.0 | 99.4 | 99.2 | 49.7 | 49.8 | 94.8 | 55.6 | 49.8 | 68.8 |
| F3Net | 50.1 | 99.9 | **99.9** | 49.9 | 50.0 | **99.9** | 49.9 | 49.9 | 68.7 |
| GramNet | 54.2 | 99.2 | 99.1 | 50.3 | 54.6 | 98.9 | 50.8 | 51.7 | 69.9 |
| DIRE | 60.2 | 99.9 | 99.8 | 50.9 | 55.0 | 99.2 | 50.1 | 50.2 | 70.7 |
| UnivFD | 73.2 | 84.2 | 84.0 | 55.2 | 76.9 | 75.6 | 56.9 | 80.3 | 73.3 |
| PatchCraft | 79.0 | 89.5 | 89.3 | 77.3 | 78.4 | 89.3 | 83.7 | 72.4 | 82.3 |
| NPR | 81.0 | 98.2 | 97.9 | 76.9 | 89.8 | 96.9 | 84.1 | 84.2 | 88.6 |
| FatFormer | 92.7 | **100.0** | 99.9 | 75.9 | 88.0 | 99.9 | **98.8** | 55.8 | 88.9 |
| GenDet | 89.6 | 96.1 | 96.1 | 58.0 | 78.4 | 92.8 | 66.5 | 75.0 | 81.6 |
| DRCT | 91.5 | 95.0 | 94.4 | 79.4 | 89.1 | 94.6 | 90.0 | 81.6 | 89.4 |
| WePe* | **91.7** | 99.5 | 98.4 | **82.3** | **93.6** | 98.1 | 95.0 | **87.1** | **93.2** |
| | | | | Training-free Methods | | | | | |
| AEROBLADE | 80.3 | **87.5** | **86.8** | 67.2 | 81.5 | **83.7** | 51.1 | 52.5 | 73.83 |
| RIGID | **81.54** | 69.5 | 68.72 | 72.35 | **84.15** | 68.57 | 78.98 | **93.02** | 78.19 |
| WePe | 79.17 | 77.8 | 75.57 | **76.07** | 79.20 | 79.00 | **90.60** | 89.27 | **80.84** |

## N SOFTWARE AND HARDWARE

We use python 3.8.16 and Pytorch 1.12.1, and several NVIDIA GeForce RTX-3090 GPU and NVIDIA GeForce RTX-4090 GPU.

## O Using natural images only to select which layers to perturb

In our experiments, we use a small set of natural images and generated images to pick the parameters that need to be perturbed. When all the generated images are not available, we can also use only the natural images to select the layers that need to be perturbed. Specifically, we first perturb each block alone and calculate the similarity of the features on the model of the natural image before and after the perturbation, as shown in Table 12. We then sort the similarity and select the blocks with the highest similarity for perturbation. As shown in Table 13, selecting the parameters to be perturbed in this way also achieves good performance and has strong robustness.

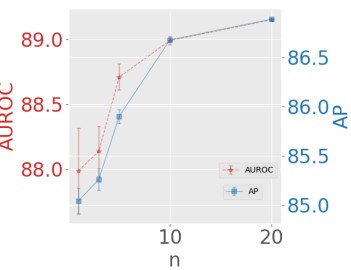

Figure 10: WePe with multiple perturbations.

## P Details of Datasets

**IMAGENET.** The natural images and generated images can be obtained at `https://github.com/layer6ai-labs/dgm-eval`. The images are provided by (Stein et al., 2023). The generative model includes: ADM, ADMG, BigGAN, DiT-XL-2, GigaGAN, LDM, StyleGAN-XL, RQ-Transformer and Mask-GIT.

Table 10: AI-generated image detection performance on LSUN-BEDROOM.

| Methods | ADM | | DDPM | | iDDPM | | Diffusion GAN | | Models Projected GAN | | StyleGAN | | Unleashing Transformer | | Average | |
|---|---|---|---|---|---|---|---|---|---|---|---|---|---|---|---|---|
| | AUROC | AP | AUROC | AP | AUROC | AP | AUROC | AP | AUROC | AP | AUROC | AP | AUROC | AP | AUROC | AP |
| CNNspot | 64.83 | 64.24 | 79.04 | 80.58 | 76.95 | 76.28 | 88.45 | 87.19 | 90.80 | 89.94 | 95.17 | 94.94 | 93.42 | 93.11 | 84.09 | 83.75 |
| Ojha | 71.26 | 70.95 | 79.26 | 78.27 | 74.80 | 73.46 | 84.56 | 82.91 | 82.00 | 78.42 | 81.22 | 78.08 | 83.58 | 83.48 | 79.53 | 77.94 |
| DIRE | 57.19 | 56.85 | 61.91 | 61.35 | 59.82 | 58.29 | 53.18 | 53.48 | 55.35 | 54.93 | 57.66 | 56.90 | 67.92 | 68.33 | 59.00 | 58.59 |
| NPR | 75.43 | 72.60 | 91.42 | 90.89 | 89.49 | 88.25 | 76.17 | 74.19 | 75.07 | 74.59 | 68.82 | 63.53 | 84.39 | 83.67 | 80.11 | 78.25 |
| WePe* | 79.41 | 76.68 | 96.71 | 96.16 | 94.18 | 93.44 | 99.81 | 99.80 | 99.83 | 99.82 | 97.06 | 96.51 | 99.45 | 99.37 | 95.21 | 94.54 |
| AEROBLADA | 57.05 | 58.37 | 61.57 | 61.49 | 59.82 | 61.06 | 47.12 | 48.25 | 45.98 | 46.15 | 45.63 | 47.06 | 59.71 | 57.34 | 53.85 | 54.25 |
| RIGID | 71.90 | 72.29 | 88.31 | 88.55 | 84.02 | 84.80 | 91.42 | 91.90 | 92.12 | 92.54 | 77.29 | 74.96 | 91.37 | 91.39 | 85.20 | 85.20 |
| WePe | 73.85 | 70.21 | 88.84 | 87.14 | 86.23 | 83.82 | 94.16 | 93.52 | 95.34 | 95.18 | 83.50 | 80.66 | 94.18 | 93.45 | 88.01 | 86.28 |

Table 11: AI-generated image detection performance (ACC, %) on DRCT-2M.

| Method | SD Variants | | | | | | Turbo Variants | | LCM Variants | | ControlNet Variants | | | DR Variants | | | Avg. |
|---|---|---|---|---|---|---|---|---|---|---|---|---|---|---|---|---|---|
| | LDM | SDv1.4 | SDv1.5 | SDv2 | SDXL | SDXL-Refiner | SD-Turbo | SDXL-Turbo | LCM-SDv1.5 | LCM-SDXL | SDv1-Ctrl | SDv2-Ctrl | SDXL-Ctrl | SDv1-DR | SDv2-DR | SDXL-DR | |
| CNNspot | 99.87 | 99.91 | 99.90 | 97.63 | 66.25 | 86.55 | 86.15 | 72.42 | 98.26 | 61.72 | 97.96 | 85.89 | 82.94 | 60.93 | 51.41 | 50.28 | 81.12 |
| F3Net | 99.85 | 99.78 | 99.79 | 88.60 | 55.85 | 87.37 | 63.29 | 63.66 | 97.39 | 54.98 | 97.98 | 72.39 | 81.99 | 65.42 | 50.39 | 50.27 | 71.13 |
| CLIP/RN50 | 99.00 | 99.99 | 99.96 | 94.61 | 62.08 | 91.43 | 84.40 | 64.40 | 98.97 | 57.43 | 99.74 | 80.69 | 82.03 | 65.83 | 50.67 | 50.47 | 80.05 |
| GramNet | 99.40 | 99.01 | 98.84 | 95.30 | 62.63 | 80.68 | 71.19 | 69.32 | 93.05 | 57.02 | 99.97 | 75.55 | 82.68 | 51.23 | 50.01 | 50.08 | 76.62 |
| De-fake | 92.10 | 95.53 | 99.51 | 89.65 | 64.02 | 69.24 | 92.00 | 93.93 | 99.13 | 70.89 | 58.98 | 62.34 | 66.66 | 50.12 | 50.16 | 50.00 | 75.52 |
| Conv-B | 99.97 | 100.0 | 99.97 | 95.84 | 64.44 | 82.00 | 60.75 | 99.27 | 99.29 | 62.33 | 99.80 | 83.40 | 73.28 | 61.65 | 51.79 | 50.41 | 79.11 |
| Ojha | 98.30 | 96.22 | 96.33 | 93.83 | 91.01 | 93.91 | 86.38 | 85.92 | 90.44 | 89.99 | 90.41 | 81.06 | 89.06 | 51.96 | 51.03 | 50.46 | 83.46 |
| DIRE | 54.62 | 75.89 | 76.04 | 99.87 | 59.90 | 93.08 | 97.55 | 87.29 | 72.53 | 67.85 | 99.69 | 64.40 | 64.40 | 49.96 | 52.48 | 49.92 | 72.55 |
| DRCT | 94.45 | 94.35 | 94.24 | 95.05 | 96.41 | 95.38 | 94.81 | 94.48 | 91.66 | 95.54 | 93.86 | 93.50 | 93.54 | 84.34 | 83.20 | 67.61 | 91.35 |
| FatFormer | 96.52 | 95.31 | 93.27 | 91.99 | 92.87 | 91.78 | 88.15 | 87.48 | 92.82 | 91.76 | 90.28 | 86.99 | 88.19 | 65.92 | 60.15 | 55.13 | 85.53 |
| WePe | 92.38 | 67.18 | 65.88 | 74.05 | 75.62 | 72.23 | 66.82 | 62.46 | 66.88 | 77.25 | 75.41 | 74.92 | 80.34 | 63.98 | 59.65 | 59.68 | 70.92 |
| WePe* | 97.06 | 96.03 | 94.76 | 96.45 | 96.59 | 97.81 | 93.54 | 92.66 | 96.29 | 94.43 | 96.69 | 96.17 | 95.72 | 75.99 | 73.32 | 69.78 | 91.45 |

**LSUN-BEDROOM.** The natural images and generated images can be obtained at `https://github.com/layer6ai-labs/dgm-eval`. The images are provided by (Stein et al., 2023). The generative model includes: ADM, DDPM, iDDPM, StyleGAN, Diffusion-Projected GAN, Projected GAN and Unleashing Transformers.

**GenImage.** The natural images and generated images can be obtained at `https://github.com/GenImage-Dataset/GenImage`. The images are provided by (Zhu et al., 2023). The generative model includes: Midjourney, SD V1.4, SD V1.5, ADM, GLIDE, Wukong, VQDM and BigGAN. The natural images come from ImageNet, and different images have different resolutions.

**DRCT-2M.** The natural images of DRCT-2M come from CoCo and can be obtained from `https://cocodataset.org/#download`. AI-generated images of DRCT-2M can be obtained from `https://modelscope.cn/datasets/BokingChen/DRCT-2M/files`, which are provided by (Chen et al., 2024). The generative model includes LDM, SDv1.4, SDv1.5, SDv2, SDXL, SDXL-Refiner, SD-Turbo, SDXL-Turbo, LCM-SDv1.5, LCM-SDXL, SDv1-Ctrl, SDv2-Ctrl, SDXL-Ctrl, SDv1-DR, SDv2-DR, SDXL-DR.

# Q   Implementation details

To balance detection performance and efficiency, we use DINOv2 ViT-L/14. We report the average results under five different random seeds and report the variance in Figure 10. In our experiments we find that perturbing the high layers may lead to a large corruption in the features of the natural images, resulting in sub-optimal results. Therefore, We do not perturb the high-level parameters. In DINOv2 ViT-L/14, the model has 24 transformer blocks, and we only perturb the parameters of the first 19 blocks with Gaussian perturbations of zero mean. The variance of Gaussian noise is proportional to the mean value of the parameters in each block, with the ratio set to 0.1. Considering the computational cost, we perturb the model only 1 time, i.e., $n = 1$. Multiple perturbations can further improve the performance as shown in Table 10. For WePe*, we leverage LoRa (Hu et al., 2022) for parameter-effcient fine-tuning. The Lora layers are applied on the q_proj and v_proj layers of DINOv2. $lora\_r$ and $lora\_\alpha$ are set to 8. And the model is optimized using the AdamW optimizer with a learning rate of $1 \times 10^{-5}$, $\beta_1 = 0.9$, $\beta_2 = 0.99$, and a weight decay of 0.01. Following CNNspot (Wang et al., 2020), data augmentation techniques including JPEG compression and Gaussian blur are employed to enhance robustness. For the IMAGENNET and LSUN-BEDROOM benchmarks, the ProGAN dataset serves as the training set. For the GenImage benchmark, SDv1.4 dataset is used as training set. For the DRCT-2M benchmark, SDv2 dataset is used as training set. When testing, to ensure objectivity in calculating classification accuracy and mitigate biases

Table 12: Effect of perturbation position on natural images. We perturb each block individually, observe the similarity of features on the model of the natural image before and after the perturbation and rank these blocks.

| block | 0 | 1 | 2 | 3 | 4 | 5 | 6 | 7 | 8 | 9 | 10 | 11 | 12 | 13 | 14 | 15 | 16 | 17 | 18 | 19 | 20 | 21 | 22 | 23 |
|---|---|---|---|---|---|---|---|---|---|---|---|---|---|---|---|---|---|---|---|---|---|---|---|---|
| similarity(%) | 99.40 | 97.66 | 98.83 | 99.00 | 98.80 | 98.70 | 99.37 | 94.73 | 92.87 | 98.44 | 97.07 | 98.00 | 93.46 | 96.24 | 94.80 | 93.85 | 92.40 | 87.60 | 71.50 | 76.00 | 80.27 | 75.93 | 34.81 | 47.90 |
| rank | 1 | 9 | 4 | 3 | 5 | 6 | 2 | 13 | 16 | 7 | 10 | 8 | 15 | 11 | 12 | 14 | 17 | 18 | 22 | 20 | 19 | 21 | 24 | 23 |

Table 13: AI-generated image detection performance on ImageNet. We select the top-k blocks with the highest similarity for perturbation based on the sorting results.

| Methods | ADM | | ADMG | | LDM | | DiT | | BigGAN | | GigaGAN | | StyleGAN XL | | RQ-Transformer | | Mask GIT | | Average | |
|---|---|---|---|---|---|---|---|---|---|---|---|---|---|---|---|---|---|---|---|---|
| | AUROC | AP | AUROC | AP | AUROC | AP | AUROC | AP | AUROC | AP | AUROC | AP | AUROC | AP | AUROC | AP | AUROC | AP | AUROC | AP |
| | | | | | | | | | Training Methods | | | | | | | | | | | |
| CNNspot | 62.25 | 63.13 | 63.28 | 62.27 | 63.16 | 64.81 | 62.85 | 61.16 | 85.71 | 84.93 | 74.85 | 71.45 | 68.41 | 68.67 | 61.83 | 62.91 | 60.98 | 61.69 | 67.04 | 66.78 |
| Ojha | 83.37 | 82.95 | 79.60 | 78.15 | 80.35 | 79.71 | 82.93 | 81.72 | 93.07 | 92.77 | 87.45 | 84.88 | 85.36 | 83.15 | 85.19 | 84.22 | 90.82 | 90.71 | 85.35 | 84.25 |
| DIRE | 51.82 | 50.29 | 53.14 | 52.96 | 52.83 | 51.84 | 54.67 | 55.10 | 51.62 | 50.83 | 50.70 | 50.27 | 50.95 | 51.36 | 55.95 | 54.83 | 52.58 | 52.10 | 52.70 | 52.18 |
| NPR | 85.68 | 80.86 | 84.34 | 79.79 | 91.98 | 86.96 | 86.15 | 81.26 | 89.73 | 84.46 | 82.21 | 78.20 | 84.13 | 78.73 | 80.21 | 73.21 | 89.61 | 84.15 | 86.00 | 80.84 |
| | | | | | | | | | Training-free Methods | | | | | | | | | | | |
| AEROBLADA | 55.61 | 54.26 | 61.57 | 56.58 | 62.67 | 60.93 | 85.88 | 87.71 | 44.36 | 45.66 | 47.39 | 48.14 | 47.28 | 48.54 | 67.05 | 67.69 | 48.05 | 48.75 | 57.87 | 57.85 |
| RIGID | 87.16 | 85.08 | 80.09 | 77.07 | 72.43 | 69.30 | 70.40 | 65.94 | 90.08 | 89.26 | 86.39 | 84.11 | 86.32 | 85.44 | 90.06 | 88.74 | 89.30 | 89.25 | 83.58 | 81.58 |
| WePe top-8 | 89.25 | 86.53 | 82.66 | 78.08 | 79.29 | 73.88 | 78.53 | 72.48 | 93.90 | 92.61 | 92.07 | 89.65 | 93.06 | 91.26 | 92.68 | 89.84 | 89.85 | 86.91 | 87.92 | 84.59 |
| WePe top-10 | 89.57 | 86.67 | 82.62 | 79.33 | 78.95 | 74.42 | 77.15 | 72.29 | 92.65 | 91.36 | 91.91 | 90.60 | 93.77 | 92.71 | 93.17 | 91.76 | 88.42 | 86.46 | 87.58 | 85.07 |
| WePe top-12 | 89.23 | 87.86 | 84.38 | 81.19 | 78.63 | 74.13 | 75.33 | 70.50 | 94.29 | 93.81 | 92.53 | 91.71 | 94.64 | 94.32 | 93.15 | 92.15 | 89.90 | 88.22 | 88.01 | 85.99 |
| WePe top-14 | 89.69 | 87.57 | 82.60 | 79.24 | 79.69 | 76.06 | 76.74 | 71.26 | 93.05 | 92.30 | 92.45 | 91.23 | 94.71 | 94.78 | 94.96 | 94.22 | 89.44 | 88.14 | 88.15 | 86.09 |
| WePe top-16 | 90.58 | 89.40 | 84.80 | 82.08 | 80.28 | 76.54 | 76.57 | 72.88 | 92.81 | 92.55 | 92.11 | 91.10 | 92.89 | 92.72 | 93.05 | 92.26 | 91.46 | 90.60 | 88.28 | 86.68 |
| WePe top-18 | 90.02 | 87.83 | 83.39 | 80.58 | 79.12 | 74.64 | 76.18 | 71.12 | 91.82 | 91.36 | 92.26 | 91.71 | 93.77 | 93.39 | 93.68 | 92.89 | 89.12 | 87.57 | 87.71 | 85.68 |

arising from manually selected thresholds, following (Ojha et al., 2023), we automatically determine the optimal threshold by identifying the score that maximizes the separation between natural and generated images, based on their computed classification scores.

