# OpenReview forum: "Epistemic Uncertainty for Generated Image Detection"
_NeurIPS.cc/2025/Conference — NeurIPS 2025 poster_

### Official Review · Reviewer_C7Yz · 2025-06-19

**Clarity:** 2
**Significance:** 2
**Originality:** 3
**Rating:** 4
**Confidence:** 2

**Summary:**

The distribution shift between training images and testing images causes higher uncertainties of machine learning models on testing compared to training data. Based on this observation, the authors propose to use a model pretrained on natural images, and compute its uncertainty on generated images, which differ in distribution. They choose a large model pretrained on a massive amount of images (DINOv2) to ensure the maximum generalizability of the approach. However, DINOv2 does not directly compute its uncertainty so the authors developed a method to do so (WePe). Then, they test the approach and its variant that additionally trains on generated images (WePe*) and compare it to other existing methods.

**Questions:**

1. Are you sure that "DINOv2 is exclusively pre-trained on natural images" (line 172)? In my opinion it is impossible to really know, and this assumption will likely be even less true with more recent models because generated images are more and more present online.

**Ethical Concerns:**

["NO or VERY MINOR ethics concerns only"]

**Final Justification:**

My weaknesses and questions were addressed in the rebuttal. Overall, the paper is interesting.

**Limitations:**

addressed in the Appendix

**Paper Formatting Concerns:**

no major concern

**Quality:**

2

**Strengths And Weaknesses:**

**Strengths**

1. Using uncertainty in the context of generated image detection is interesting and opens the gate to reuse many methods of uncertainty quantification and out-of-distribution detection for this application.
1. Well-justified method of uncertainty estimation (WePe) to obtain the uncertainty of DINOv2, and it includes a variant benefitting from training (WePe*).
1. Experimental results on many generative models and with many competing methods on four datasets.

**Weaknesses**

1. Lack of standard uncertainty quantification baselines. For instance, the authors could use a pretrained ImageNet classifier and use the maximum predicted probability (or logit), or the entropy as a confidence indicator. Without this, it is hard to know whether the WePe method based on weight perturbation is necessary.
1. Not enough details on the data (number of samples, splits used...) for the results and figures (e.g., for Figures 1, 2, 3, where are the natural and fake images coming from?). If the natural images for ImageNet include the training data, then all other images would have very different features, even for natural images from validation set (because DINOv2 was trained on ImageNet train set). Because of the assumptions made, it is necessary to provide such details.

---

> ### Author Rebuttal · Authors · 2025-07-31
>
> We thank the reviewer for the thorough review and constructive feedback. We address each concern below.
>
> > Q1. Lack of standard uncertainty quantification baselines. For instance, the authors could use a pretrained ImageNet classifier and use the maximum predicted probability (or logit), or the entropy as a confidence indicator. Without this, it is hard to know whether the WePe method based on weight perturbation is necessary.
>
> We express our gratitude for the reviewer’s insightful suggestion. Two primary considerations preclude the adoption of conventional metrics for assessing confidence levels in our study:
>
> - In the context of detecting AI-generated images, image categories is diverse and inherently open-ended. Classifiers trained on the ImageNet dataset—which encompasses a finite set of categories—can only yield reliable confidence scores for in-distribution categories. For images beyond this taxonomy—whether natural or generated—the confidence scores produced are unreliable.
>
> - Standard image classifiers generate confidence scores that reflect their confidence in predicting object categories. However, the distinction between natural and generated images lies not in categorical differences but in attributes such as texture and structural patterns. Consequently, we opted for WePe over classifier outputs to estimate uncertainty.
>
> To address the reviewer’s concerns, we conducted experiments using a ResNet18 model pretrained on ImageNet to differentiate between natural and generated images, employing maximum softmax probability (MSP) and entropy as confidence metrics. As demonstrated in the table below, these metrics exhibit suboptimal performance.
>
>
> |         | AUROC | AP    |
> |---------|-------|-------|
> | MSP     | 48.85 | 49.23 |
> | entropy | 51.76 | 50.28 |
> | WePe    | 87.99 | 85.04 |
>
> Thanks again for your insightful question. We have added the above explanations to our revision.
>
>
> > Q2. Not enough details on the data for the results and figures. If the natural images for ImageNet include the training data, then all other images would have very different features, even for natural images from validation set (because DINOv2 was trained on ImageNet train set). Because of the assumptions made, it is necessary to provide such details.
>
> We express our gratitude for the reviewer’s valuable suggestion. For Figures 1 and 3, natural images are sourced from the LSUN dataset, and generated images are produced using the ProGAN model. For Figure 3, natural images are sampled from the LSUN-Bedroom dataset, with generated images created by the following models: ADM, DDPM, iDDPM, StyleGAN, Diffusion-Projected GAN, Projected GAN, and Unleashing Transformers.
>
>
> > Q3. Are you sure that "DINOv2 is exclusively pre-trained on natural images" (line 172)? In my opinion it is impossible to really know, and this assumption will likely be even less true with more recent models because generated images are more and more present online.
>
>
> Thanks for pointing out this potentially confusing statement. The original statement is overly absolute. The DINOv2 model leverages multiple curated datasets composed of natural images. Additionally, it incorporates data sourced from the internet that aligns with these curated datasets, potentially introducing some AI-generated images. However, due to the known issue of mode collapse [1], few foundational models intentionally include a substantial proportion of generated images during training. Consequently, generated images remain a long-tail component relative to natural images. Nevertheless, as generative models advance, the increasing prevalence of generated images online may contaminate training datasets for new models. To address this, we propose precise calibration to effectively widen the uncertainty gap between natural and generated images. The efficacy of this approach is substantiated by our experimental results.
>
>
> Reference:
>
> [1]: AI models collapse when trained on recursively generated data. Shumailov et al. Nature 2024

---

> > ### Comment · Reviewer_C7Yz · 2025-08-02
> > **Response to rebuttal**
> >
> > Thank you for this detailed rebuttal.
> >
> > Q1 follow-up. I appreciate the baseline results. However it would be a more fair comparison to use a better model than ResNet18, for instance Vit-L/14 (the same architecture as DINOv2). In this case, the difference would be the training data and training method, but not the architecture/model capacity (which might have an impact or not). I agree with the two primary considerations you wrote, but it does not preclude from including the results in the table which in my opinion allow for better understanding of the quality of the results.
> >
> > Q2 follow-up. Sorry my comment was maybe unclear. Let me reformulate: I asked whether the natural images were sampled from the train split or the test split.

---

> > > ### Author Response · Authors · 2025-08-03
> > >
> > > Thank you for your prompt reply and constructive comments! Please find our responses below.
> > >
> > >
> > > Q1. However it would be a more fair comparison to use a better model than ResNet18, for instance Vit-L/14 (the same architecture as DINOv2). In this case, the difference would be the training data and training method, but not the architecture/model capacity (which might have an impact or not).
> > >
> > > We appreciate your suggestion. To eliminate the influence of model architecture, we additionally conducted experiments using the pre-trained ViT-L/14 model. The results are shown in the following table. We have included these results and the corresponding discussion in the revised version to provide a fairer comparison and a clearer evaluation of method performance.
> > >
> > > |         | AUROC | AP    |
> > > |---------|-------|-------|
> > > | MSP     | 63.23 | 60.16 |
> > > | entropy | 65.79 | 61.97 |
> > > | WePe    | 87.99 | 85.04 |
> > >
> > >
> > > Q2. Sorry my comment was maybe unclear. Let me reformulate: I asked whether the natural images were sampled from the train split or the test split.
> > >
> > > Thank you for the clarification. The natural images used in our experiments were sampled from the test split. We have revised the manuscript accordingly to make this point explicit.

---

> > > > ### Comment · Reviewer_C7Yz · 2025-08-04
> > > > **Discussion**
> > > >
> > > > Thank you for your effort. The baseline results are reasonably good, and are interesting to include in the table (it even outperforms some of the other methods). I will increase my rating.

---

> > > > > ### Author Response · Authors · 2025-08-04
> > > > >
> > > > > Thank you for your swift response, especially during such a busy period. We sincerely appreciate that you can raise the score.  We have incorporated the above discussion and results into the revised version. If there are any remaining concerns or suggestions for improvement, we would be happy to address them further.

---

### Official Review · Reviewer_4Yn5 · 2025-06-25

**Clarity:** 4
**Significance:** 3
**Originality:** 2
**Rating:** 5
**Confidence:** 4

**Summary:**

This papers tackles the problem of AI-generated image detection through a novel approach named WePe (weight perturbation). In contrast to other work where models are trained to distinguish between real and generated images, this work focuses on the distribution shift between the real distribution and the artificial one to capture the epistemic uncertainty of the model under parameter perturbation. The main supposition is that, since the used models are trained on the original dataset images, a perturbation on its weights will show greater sensitivity to generated images than to original ones. This new detection system has the advantage of not relying on generated images during training, thus it simplifies the process of dataset selection and pre-processing techniques.
A calibrated approached through a regularization loss with access to generated images is also presented and named WePe*.
Authors perform experiments on 4 datasets and compare 3 metrics with a wide set of baselines for training and training-free methods.

**Questions:**

**Q1** - Given that epistemic uncertainty captures the lack of knowledge about a distribution, wouldn’t it be beneficial to also consider aleatoric uncertainty—especially if the generated distribution is misaligned with the natural one? Additionally, should domain uncertainty (often considered a third type of uncertainty related to OOD data) be taken into account in this context?


**Q2** - In Equation (13), since the expression is an upper bound, could the authors elaborate on its implications? In particular, how do you ensure (or to what extent can we expect) that it provides a reliable basis for ranking the instances?

**Ethical Concerns:**

["NO or VERY MINOR ethics concerns only"]

**Final Justification:**

The paper is solid, and the authors addressed the main concerns. I’m raising my score to accept.

**Limitations:**

Yes.

**Paper Formatting Concerns:**

No.

**Quality:**

3

**Strengths And Weaknesses:**

**Strengths:**

**S1** – The paper is clearly written, with a well-articulated motivation and a solid introduction to the related work and background. Overall, it is easy to follow.


**S2** – The proposed technique works even if no generated images are available for training. This makes the training easier and the technique scalable.

**S3** – Mathematical claims seem to be correct. Epistemic uncertainty is correctly approached through model weight perturbation.

**S4** – Even if there is no access to the teacher model’s parameters, the upper bound on the uncertainty allows for an effective estimation of uncertainty.

**S5** – Metrics are well chosen. Fréchet Inception Distance (FID) is the standard metric to compare the distribution of generated images to the real distribution. AUROC and Average Precision give a global view of the performance of a model. Perturbation Sensitivity also properly reflects the effect of variations in the parameters of the model and introduces well Theorem 3.2.

**S6** – Wide range of experiments, datasets, and baselines. The ablation study further confirms the validity of the results.

**S7** – Results show that the proposed method works well on detecting generated images, as well as achieving higher robustness to attacks than the other methods.

**S8** – Some extra conclusions taken from Section 4.3 provide additional insights—for example, the study on which layers to perturb or the degree of perturbation.

**Weaknesses:**

**W1 –** Some important results are delegated to the Appendix, which may hinder the overall flow of the paper. If the authors believe it could improve readability, they might consider restructuring certain sections so that key findings are more accessible in the main text, reducing the need to jump frequently to the supplementary material.

**W2 –** It is not entirely clear what statistical analysis has been carried out on the reported results, and the paper lacks confidence intervals in the main tables. In the *Experimental Details* paragraph on page 7, the authors mention averaging results over 5 random seeds, but it is unclear what specific results this refers to. Figure 9 is then introduced (in the Appendix), but its meaning and interpretation are not clearly explained. From what we understand, Figure 9 shows variability across 5 runs, but only for the proposed method. Has a similar analysis been conducted for the other methods used in the benchmarking? Strengthening this point in the Appendix and including confidence intervals for all compared methods when available, would improve the overall quality of the experimental findings.

**W3 –** The related work on Uncertainty Quantification feels somewhat limited in scope. It could be enriched by including references more directly connected to how uncertainty is measured in this paper. If the authors believe such additions would be relevant, this could help better position their contribution in the context of existing literature.

**Some minor comments:**

- **C1: Typos and small formatting issues**
    - Line 79: “We” should start with lowercase after the comma.
    - Lines 86 and 87: The word “discrepancy” is missing the final “y” (appears twice).
    - Line 180: The word “However” contains a typo.
    - Line 241: The method SelfBlend is misspelled as SelfBland.
    - Table 1: AEROBLADE is written incorrectly as AEROBLADA.
    - Line 250: “We” should be lowercase after the comma.
    - Line 294: “Naturak” should be corrected to “natural”.
    - Line 563: “wepe” should be capitalized as “WePe”.
- **C2: Possible notation inconsistencies:**
    - Section 2.2: There is a notation inconsistency regarding the model’s weights. Are they written as $\theta$ or $\textbf{w}$? Do these notations express different things?
    - Theorem 3.2 vs. Section 2.1: The distributions for AI-generated and natural images are $\mathcal{D}^0$ and $\mathcal{D}^1$ in the latter, and the other way around in the former, and again recovered in Equation (16).
    - Line 261: You mention that when WePe is “further trained to amplify the…” but after the comma you name it WePe instead of WePe*, which is the notation you adopted for this method. This can lead to confusion.
- **C3: Clarity improvements**
    - Table 1: Under *Training Methods*, the methods — PatchCraft, FatFormer, and DRCT — are listed, but none of them are mentioned or described in the main text of the paper. Including brief descriptions or citations would improve clarity and completeness.
    - Section 4.3 (first paragraph title): Consider renaming the paragraph title to “Robustness to Image Perturbations” to clarify that the perturbations refer to input images rather than model weights.

**C4: Results consistency.** Some of the results presented in Table 4 appear to be incorrectly computed, if we understood the methodology correctly. The difference in AUROC seems to be calculated based on Table 1 results, but the numbers for CNNspot, NPR, and WePe* do not match. For example, for CNNspot: 59.78−67.04=−7.2659.78 - 67.04 = -7.2659.78−67.04=−7.26, not −9.06-9.06−9.06 as stated.

---

> ### Author Rebuttal · Authors · 2025-07-31
>
> We thank the reviewer for the thorough review and constructive feedback. We address each concern below.
>
> > Q1. Some important results are delegated to the Appendix, which may hinder the overall flow of the paper.
>
> Thank you for your kind suggestion. We agree with your point, and, in the revised version, we have reorganized the tables to enhance the clarity and flow of the paper.
>
> > Q2. It is not entirely clear what statistical analysis has been carried out on the reported results, and the paper lacks confidence intervals in the main tables.
>
> Thanks for your constructive suggestion. Since the perturbations are random, the results of our method are the average values under five random seeds. In Figure 9, we report the error bars of our method. For clarity, we have added this information to the revised table. For other methods, since some of the results were obtained directly from related papers, we did not report this information.
>
>
> > Q3. The related work on Uncertainty Quantification feels somewhat limited in scope.
>
> In response to your valuable suggestion, we have expanded the discussion on uncertainty estimation in the related work section to provide a more comprehensive and rigorous analysis.
>
> Blundell et al. (2015) introduce Bayes by Backprop, a method for estimating weight uncertainty in neural networks by modeling the posterior distribution over weights using variational inference, improving model generalization and robustness. Similarly, Ferrante et al. (2024) leverage weight perturbation techniques to estimate neural network uncertainty, demonstrating improved classification accuracy through robust uncertainty quantification. Pearce et al. (2018) explore distribution-free uncertainty estimation, using conformal prediction and quantile regression to estimate bounds on aleatoric uncertainty. Chan et al. (2025) introduce hyper-diffusion models, allowing to accurately estimate both epistemic and aleatoric uncertainty with a single model.
>
>
> > Q4. Comments on typos.
>
> Thank you for your careful review! We have carefully revised all typos.
>
> > Q5. Possible notation inconsistencies and results inconsistencies.
>
> Thank you for pointing out these issues! We have revised these inconsistencies in the revised version.
>
> > Q6. Clarity improvements.
>
> We have incorporated a discussion on PatchCraft, FatFormer, and DRCT in the related work section and revised the title from "Robustness to Perturbations" to "Robustness to Image Perturbations" for greater clarity.
>
>
> > Q7. Given that epistemic uncertainty captures the lack of knowledge about a distribution, wouldn’t it be beneficial to also consider aleatoric uncertainty—especially if the generated distribution is misaligned with the natural one? Additionally, should domain uncertainty (often considered a third type of uncertainty related to OOD data) be taken into account in this context?
>
>
> We appreciate your insightful feedback. Aleatoric uncertainty quantifies the inherent noise within observational data. Given the distinct data distributions of natural and generated images, their aleatoric uncertainties may differ, potentially serving as a discriminative feature to distinguish between them. Domain uncertainty, arising from shifts in data distribution, may exhibit effects comparable to those of epistemic uncertainty.
>
> > Q8. In Equation (13), since the expression is an upper bound, could the authors elaborate on its implications? In particular, how do you ensure (or to what extent can we expect) that it provides a reliable basis for ranking the instances?
>
>
> In formula 13, the inequality is obtained through the Cauchy-Schwarz inequality, and we can explore the validity of the upper bound by calculating the difference between the two sides of the inequality. We define the difference as:
>
> $$ \triangle = ||f\left(\mathbf{x} ; \theta_k\right)-\frac{1}{n} \sum_j^n f\left(\mathbf{x} ; \theta_j\right)||^2  \cdot ||f\left(\mathbf{x} ; \theta_t\right)||^2 - ||(f\left(\mathbf{x} ; \theta_k\right)-\frac{1}{n} \sum_j^n f\left(\mathbf{x} ; \theta_j\right)) \cdot (f\left(\mathbf{x} ; \theta_t\right))||^2$$
>
>
> Thus, the difference $\triangle$ can be written as:
>
> $$
> \triangle = \left\| f\left(\mathbf{x}; \theta_k\right) - \frac{1}{n} \sum_{j=1}^n f\left(\mathbf{x}; \theta_j\right) \right\|^2 \cdot \left\| f\left(\mathbf{x}; \theta_t\right) \right\|^2 - \left( \left\| f\left(\mathbf{x}; \theta_k\right) - \frac{1}{n} \sum_{j=1}^n f\left(\mathbf{x}; \theta_j\right) \right\| \left\| f\left(\mathbf{x}; \theta_t\right) \right\| \cos \theta \right)^2
> $$
>
> where $\theta$ is the angle between $f\left(\mathbf{x} ; \theta_t\right)$ and $f\left(\mathbf{x} ; \theta_k\right)-\frac{1}{n} \sum_j^n f\left(\mathbf{x} ; \theta_j\right)$.
>
> And finally, we obtain the following expression:
>
> $$
> \triangle = \left\| f\left(\mathbf{x}; \theta_k\right) - \frac{1}{n} \sum_{j=1}^n f\left(\mathbf{x}; \theta_j\right) \right\|^2 \cdot \left\| f\left(\mathbf{x}; \theta_t\right) \right\|^2 \left( 1 - \cos^2 \theta \right)
> $$
> Using the trigonometric identity, we have:
> $$
> \triangle = \left\| f\left(\mathbf{x}; \theta_k\right) - \frac{1}{n} \sum_{j=1}^n f\left(\mathbf{x}; \theta_j\right) \right\|^2 \cdot \left\| f\left(\mathbf{x}; \theta_t\right) \right\|^2 \sin^2 \theta
> $$
>
> This result indicates that the smaller the angle between $f\left(\mathbf{x} ; \theta_k\right)-\frac{1}{n} \sum_j^n f\left(\mathbf{x} ; \theta_j\right)$ and $f\left(\mathbf{x} ; \theta_t\right)$, the tighter the upper bound. In our experiment, the perturbation applied to the model is extremely small, causing the image features to remain virtually unchanged, resulting in an extremely weak deviation between $f\left(\mathbf{x} ; \theta_k\right)-\frac{1}{n} \sum_j^n f\left(\mathbf{x} ; \theta_j\right)$ and $f\left(\mathbf{x} ; \theta_t\right)$, thereby enabling the upper bound to provide a reliable basis for ranking the instances.

---

> ### Comment · Reviewer_4Yn5 · 2025-08-06
>
> I appreciate the effort the authors made in addressing my review. The revisions regarding formatting, typos, and inconsistencies in notation and results (Q1, Q4, Q5, Q6) are welcome, and I believe the paper is now more readable.
>
> However, I still have concerns about the statistical analysis presented, particularly in relation to Figure 9 (Q2). While the rebuttal mentions that the figure shows error bars derived from five runs of your method, it remains unclear what exactly is being represented. I attempted to reconcile the means shown in the tables with the points in the figure, but I was unable to clearly understand the correspondence. Additionally, the source and nature of the reported variance are not sufficiently explained. I understand that for baseline methods taken from related literature it may not be possible to include variance, but for your own method, it would be important to show the error bars for all metrics wherever applicable. Ideally, this should be done not only in graphical form (as in Figure 9), but also explicitly in tabular form, so the reader can interpret the results with more confidence. A clarification of what exactly is shown in Figure 9 and how it aligns with the rest of the results would be very helpful.
>
> The additional references you included on UQ (Q3) are more adequate to the scope of the paper. These are references presenting methods that work on the model’s weights, leveraging their variations to measure uncertainties. This motivates better your method.
>
> Regarding the answer to Q7 on aleatoric and domain uncertainty, while the distinction between types of uncertainty is well outlined and their sources are acknowledged, the response does not elaborate on how these forms of uncertainty could be incorporated into the proposed method.
>
> Finally, the response to the question on the upper bound of uncertainty (Q8) is sufficiently rigorous and resolves the issue raised. I would suggest including this derivation in the Appendix so that readers can follow the reasoning in detail.
>
> I think this a good paper and, in general, the rebuttal addressed my concerns. I will increase my score to accept.

---

> > ### Author Response · Authors · 2025-08-06
> >
> > We sincerely thank the reviewer for the supportive feedback. Below, we provide a clearer explanation of your concerns.
> >
> > In Figure 9, we present two observations. First, performance can be further improved by applying multiple perturbations to the model. For example, when $n=1$—corresponding to the performance reported in Table 1—the AUROC is 87.99%. When $n=20$, the model is randomly perturbed 20 times and the average feature similarity between the original model and these perturbed models is computed; in this setting, the AUROC exceeds 89%. Second, we report error bars for different perturbation counts. Since the added noise is random, different seeds can lead to slight performance variations. For $n=1$, the specific deviations are shown in the table below, which has been included in the revised manuscript.
> >
> >
> > |        | ADM   |       | ADMG  |       | LDM   |       | DiT   |       | BigGAN |       | GigaGAN |       | StyleGAN XL |       | RQ-Transformer |       | Mask GIT |       | Average |       |
> > |--------|-------|-------|-------|-------|-------|-------|-------|-------|--------|-------|---------|-------|-------------|-------|----------------|-------|----------|-------|---------|-------|
> > |        | AUROC | AP    | AUROC | AP    | AUROC | AP    | AUROC | AP    | AUROC  | AP    | AUROC   | AP    | AUROC       | AP    | AUROC          | AP    | AUROC    | AP    | AUROC   | AP    |
> > | WePe   | 89.79 | 87.32 | 83.20 | 78.80 | 78.47 | 73.50 | 77.13 | 71.21 | 94.24  | 93.64 | 92.15   | 90.29 | 93.86       | 92.86 | 93.50          | 91.47 | 89.55    | 86.25 | 87.99   | 85.04 |
> > | $\Delta$ | 0.32  | 0.18  | 0.35  | 0.24  | 0.37  | 0.42  | 0.29  | 0.11  | 0.47   | 0.24  | 0.20    | 0.26  | 0.39        | 0.25  | 0.33           | 0.22  | 0.52     | 0.23  | 0.33    | 0.14  |
> >
> >
> > Regarding aleatoric uncertainty, generated images are typically sampled from a generative model to maximize likelihood, resulting in samples concentrated in high-probability regions and therefore exhibiting lower aleatoric uncertainty. In contrast, natural images tend to have higher aleatoric uncertainty due to factors such as variations in lighting and sensor noise. Conversely, generated images show higher epistemic uncertainty on DINOv2 due to its pre-training on natural images. Leveraging the difference between epistemic and aleatoric uncertainty could enhance discrimination between natural and generated images, though this requires further validation.
> >
> > Nonetheless, estimating aleatoric uncertainty remains challenging. Classical methods [1] require modifying the network’s loss function to predict both outputs and associated variances, which is non-trivial to implement in the DINOv2 framework. Alternative approaches, such as test-time data augmentation [2,3], are also unsuitable in our case: since DINOv2 is trained predominantly on natural images, the estimated aleatoric uncertainty for generated images becomes biased and unreliable.
> >
> > To enable a fair and accurate comparison of aleatoric uncertainty between natural and generated images, a potential solution would be to train a model on a balanced dataset containing substantial numbers of both types. This could mitigate bias in the estimation and provide a more reliable basis for comparison.
> >
> > Regarding domain uncertainty, there is currently limited research on this topic, and it may share similar properties with epistemic uncertainty.
> >
> >
> >
> > We sincerely thank the reviewer for these valuable comments, which have significantly improved the paper. We have incorporated the above clarifications and results into the revised manuscript.
> >
> >
> > Reference:
> >
> > [1]: What Uncertainties Do We Need in Bayesian Deep Learning for Computer Vision?
> >
> > [2]: Test-time Data Augmentation for Estimation of Heteroscedastic Aleatoric Uncertainty in Deep Neural Networks
> >
> > [3]: Aleatoric uncertainty estimation with test-time augmentation for medical image segmentation with convolutional neural networks

---

### Official Review · Reviewer_Q2AQ · 2025-07-01

**Clarity:** 3
**Significance:** 3
**Originality:** 3
**Rating:** 4
**Confidence:** 3

**Summary:**

This paper proposes a novel method for detecting AI-generated images based on epistemic uncertainty. The key idea is to exploit the distributional shift between natural and generated images by measuring how sensitive image features are to weight perturbations in a large pre-trained vision model (e.g., DINOv2). The method, termed WePe, operates without needing generated image samples for training and can be optionally enhanced (WePe*) by fine-tuning to amplify the uncertainty gap. Experiments on multiple benchmarks demonstrate state-of-the-art detection performance among both training-based and training-free approaches.

**Questions:**

1. Could you elaborate on how sensitive your method is to the choice of the pre-trained model? While DINOv2 was shown to work well, have you explored other large-scale vision models (e.g., more recent multimodal models or domain-adapted models)?

2. The method assumes that generated images will continue to exhibit higher epistemic uncertainty. As generative models become more advanced, how do you envision addressing scenarios where this uncertainty gap narrows?

3. Could you provide more insights (e.g., failure case examples) to illustrate how the weight perturbation affects the feature space for natural vs. generated images?

4. Regarding computational cost, the author mention that WePe requires two forward passes. Could you clarify whether any optimizations (e.g., batching, parallelization) were applied in your reported efficiency measurements, and how the method scales to larger datasets?

5. Have you considered whether the proposed weight perturbation technique could be applied dynamically (e.g., with adaptive perturbation magnitudes) or combined with input-space perturbations for further performance gains?

**Ethical Concerns:**

["NO or VERY MINOR ethics concerns only"]

**Final Justification:**

Thanks for the rebuttal, most of my concerns are addressed.

**Limitations:**

yes

**Quality:**

3

**Strengths And Weaknesses:**

Strengths
1. The paper tackles an important and timely problem—AI-generated image detection with strong generalization to unseen generators—which is highly relevant given the rapid development of generative models.
2. The idea of linking epistemic uncertainty (via weight perturbation) to distribution shift is conceptually sound and supported by theoretical analysis and empirical results.
3. The method is simple, training-free in its basic form (WePe), and efficient, making it appealing for practical deployment.
4. Extensive experiments across several benchmarks and perturbation scenarios (e.g., frequency/spatial domain attacks) demonstrate competitive or superior performance relative to strong baselines.
5. The paper is clearly written, well-organized, and easy to follow.

Weaknesses:
1. The novelty is incremental in terms of core methodological innovation. While combining uncertainty estimation with weight perturbation for this task is insightful, it builds on well-known ideas from OOD detection and uncertainty quantification without introducing fundamentally new learning paradigms.
2. The method relies heavily on the assumption that future generative models will still induce detectable uncertainty gaps; while acknowledged by the authors, this limitation poses a potential risk to long-term applicability.
3. The experiments could include broader comparison with more recent detection frameworks or more diverse pre-trained models (e.g., stronger multimodal models beyond CLIP and BLIP).

---

> ### Author Rebuttal · Authors · 2025-07-31
>
> We thank the reviewer for the thorough review and constructive feedback. We address each concern below.
>
> > Q1. The novelty is incremental in terms of core methodological innovation. While combining uncertainty estimation with weight perturbation for this task is insightful, it builds on well-known ideas from OOD detection and uncertainty quantification without introducing fundamentally new learning paradigms.
>
>
> To address your concern, we have added the following clarifications in our revised paper, aiming to highlighting the distinct contributions of our work compared to OOD detection and uncertainty quantification methods:
>
> - **Contrast with OOD Detection:** Conventional OOD detection [1,2] usually relies on Maximum Softmax Probability (MSP) scores, exploiting fixed ID categories to identify OOD samples with low probabilities. In contrast, AI-generated detection involves diverse, unbounded categories, rendering MSP scores ineffective. WePe introduces a novel uncertainty estimation approach tailored for detecting generated images, achieving robust performance where traditional OOD methods falter.
>
> - **Contrast with Uncertainty Quantification:** Standard techniques like MC-Dropout and DeepEnsemble are ill-suited for DINOv2. Training multiple DINOv2 models for DeepEnsemble is computationally infeasible, and the absence of dropout in DINOv2 undermines MC-Dropout’s efficacy. Our proposed weight perturbation method overcomes these limitations, delivering a practical and effective uncertainty estimation tailored to DINOv2’s architecture, as validated by our experiments.
>
> These advancements position our approach as a pioneering solution, adeptly addressing the unique demands of AI-generated image detection with superior adaptability and performance.
>
>
> > Q2. The method relies heavily on the assumption that future generative models will still induce detectable uncertainty gaps; while acknowledged by the authors, this limitation poses a potential risk to long-term applicability.
>
> We sincerely appreciate your insightful feedback. We acknowledge the potential limitation in assuming that future generative models will consistently produce detectable uncertainty gaps. However, this assumption lies in the fact that uncertainty approaches can detect the distribution shifts. Thus, these approaches are expected to detect the distribution discrepancy between natural and generated images. The potential failure case would be that future models generate images causing little distribution shift from natural images. In this context, it is hard to ensure whether human beings can identify these generated images.
>
> > Q3. The experiments could include broader comparison with more recent detection frameworks or more diverse pre-trained models (e.g., stronger multimodal models beyond CLIP and BLIP).
>
> We appreciate your kind suggestion. To further validate the robustness of WePe, we evaluated its performance using advanced multimodal models, namely SigLIP-So400m-patch16-256 [3] and InternViT-6B-224px [4]. As presented in the table below, WePe demonstrates strong performance across these models, affirming its effectiveness and generalizability.
>
> |                        | AUROC | AP    |
> |------------------------|-------|-------|
> |DINOv2 |87.99|85.04|
> | SigLIP2                | 85.63 | 84.31 |
> | InternViT |    90.24    | 88.37     |
>
>
> > Q4. Could you provide more insights (e.g., failure case examples) to illustrate how the weight perturbation affects the feature space for natural vs. generated images?
>
> We appreciate your insightful suggestion. WePe leverages the differential epistemic uncertainty exhibited by pre-trained models, such as DINOv2, when processing natural versus generated images to differentiate between them. Specifically, DINOv2, having been trained on an extensive dataset of natural images, demonstrates lower epistemic uncertainty for such images. Images that diverge from the distribution of the training dataset—despite not being generated by a generative model—tend to elicit higher uncertainty from DINOv2, leading to their misclassification as generated images. In the revised manuscript, we visualized the feature shifts in DINOv2’s representations for a set of cartoon images, which were not produced by generative models, before and after weight perturbation. These images, due to their deviation from the training natural distribution, exhibited significant feature shifts post-perturbation, rendering them indistinguishable from generated images in our analysis.
>
>
> > Q5. Regarding computational cost, the author mention that WePe requires two forward passes. Could you clarify whether any optimizations (e.g., batching, parallelization) were applied in your reported efficiency measurements, and how the method scales to larger datasets?
>
> We appreciate your kind suggestion. To evaluate computational efficiency, we adhered to standard testing protocols, utilizing a dataloader for parallel computation. For each batch of samples, we first extract features using the original model, followed by feature extraction with the perturbed model, necessitating two forward passes on a single GPU. For large-scale datasets, we can employ two GPUs: one to compute features from the original model and another for the perturbed model. This configuration enables the acquisition of prediction results for each sample within the duration of a single forward pass, enhancing computational efficiency.
>
> > Q6. Have you considered whether the proposed weight perturbation technique could be applied dynamically (e.g., with adaptive perturbation magnitudes) or combined with input-space perturbations for further performance gains?
>
>
> Thank you for your inspiring question. Implementing adaptive perturbation may necessitate training an additional perceptron to determine the appropriate perturbation amplitude, which would incur additional computational overhead. As an alternative, we propose perturbing the model multiple times with varying amplitudes and computing the average feature values to approximate the effect of adaptive perturbation. This approach, as demonstrated in the table below, enhances the performance of WePe. Furthermore, integrating this method with input perturbation also improves WePe's performance.
>
> |                        | AUROC | AP    |
> |------------------------|-------|-------|
> |WePe|87.99|85.04|
> | WePe + multi random perturbation              | 89.36 | 90.62 |
> | WePe + input perturbation |    88.43   | 86.88      |
>
> Reference:
>
> [1]: DICE: Leveraging Sparsification for Out-of-Distribution Detection. Sun et al. ECCV 2022
>
> [2]: Extremely Simple Activation Shaping for Out-of-Distribution Detection. Djurisic et al. ICLR 2023
>
> [3]: SigLIP 2: Multilingual Vision-Language Encoders with Improved Semantic Understanding, Localization, and Dense Features. Tschannen et al. arXiv 2025
>
> [4]: Internvl: Scaling up vision foundation models and aligning for generic visual-linguistic tasks. Chen et al. CVPR 2024

---

> > ### Comment · Reviewer_Q2AQ · 2025-08-08
> > **Official Comments**
> >
> > Thanks for the rebuttal and the experiments; most of my concerns are addressed. I will keep my positive score of borderline accept.

---

### Official Review · Reviewer_frxx · 2025-07-03

**Clarity:** 3
**Significance:** 3
**Originality:** 3
**Rating:** 4
**Confidence:** 3

**Summary:**

This paper presents Weight Perturbation, a novel framework for AI-generated image detection that leverages epistemic uncertainty estimation. The core insight is that vision models pre-trained exclusively on natural images exhibit higher epistemic uncertainty when processing AI-generated images due to distributional shifts. The authors implement this through weight perturbation of pre-trained DINOv2 models, measuring feature sensitivity to these perturbations. Natural images maintain consistent features under weight perturbation, while generated images show significant variations. The approach offers both training-free weight perturbation and fine-tuned variants, demonstrating competitive performance across multiple benchmarks.

**Questions:**

1. How does WePe differentiate between epistemic uncertainty arising from AI generation versus uncertainty from other natural distributional shifts (e.g., domain changes, resolution variations, or color-space differences)? Could you discuss potential strategies to mitigate confusion with non-generative OOD samples?
2. If an attacker has white‐box access, they could conceivably craft images whose feature representations remain insensitive to weight perturbations, defeating the core detection signal. What defenses might mitigate such targeted attacks?

**Ethical Concerns:**

["NO or VERY MINOR ethics concerns only"]

**Final Justification:**

My weaknesses and questions were addressed in the rebuttal.

**Limitations:**

Yes.

**Paper Formatting Concerns:**

None.

**Quality:**

3

**Strengths And Weaknesses:**

## Strength

1. The work introduces a well-motivated conceptual framework that reframes AI-generated image detection as an OOD problem solvable through epistemic uncertainty. This departure from traditional binary classification approaches is both theoretically grounded and practically compelling.
2. The authors demonstrate SOTA performance across four challenging benchmarks, with WePe* consistently ranking among top performers and the training-free WePe remaining highly competitive. Particularly impressive is the method's demonstrated robustness to various attacks and image degradations, where competing methods show substantial performance drops while WePe maintains stability.
3. The paper provides thorough ablation studies covering perturbation types, layer selection strategies, and hyperparameter sensitivity.

## Weaknesses

1. The epistemic uncertainty-based approach faces an inherent limitation in distinguishing AI-generated images from other types of OOD samples that could naturally arise in practical deployment scenarios. Since the method fundamentally relies on detecting distributional shifts from the training data of pre-trained models, it may exhibit elevated uncertainty for legitimate natural images that differ from the training distribution due to factors such as varying image resolutions, color-space conversions, or domain shifts. This could lead to false positive detections where genuine natural images are incorrectly classified as AI-generated. The performance variations observed in the appendix results on GenImage and DRCT-2M datasets, which contain images with diverse resolutions and sources, appear to support this concern. The method lacks mechanisms to differentiate between uncertainty arising from AI generation versus uncertainty from other distributional mismatches, potentially limiting its reliability in real-world applications where input images may come from varied and unpredictable sources.
2. The method's success appears to be reliant on the specific properties of DINOv2. The paper would be more impactful if it provided a deeper analysis into *which* properties of DINOv2 are crucial for this task to establish a more general principle.

---

> ### Author Rebuttal · Authors · 2025-07-31
>
> We thank the reviewer for the thorough review and constructive feedback. We address each concern below.
>
>
> > Q1. Since the method fundamentally relies on detecting distributional shifts from the training data of pre-trained models, it may exhibit elevated uncertainty for legitimate natural images that differ from the training distribution due to factors such as varying image resolutions, color-space conversions, or domain shifts. This could lead to false positive detections where genuine natural images are incorrectly classified as AI-generated. The method lacks mechanisms to differentiate between uncertainty arising from AI generation versus uncertainty from other distributional mismatches, potentially limiting its reliability in real-world applications where input images may come from varied and unpredictable sources.
>
> We appreciate your insightful comments. WePe leverages heightened epistemic uncertainty arising from distribution shifts to differentiate between natural and AI-generated images. However, natural images that diverge from the training distribution may be misclassified as generated. To mitigate this challenge, we made two efforts in our paper:
>
> - We select DINOv2 as the feature extractor, as its training on a diverse and extensive dataset of natural images ensures a broad and inclusive training distribution, encompassing a wide range of natural image variations.
>
> - We further amplify the uncertainty induced by AI generation through confidence calibration, enhancing the robustness of our detection approach. Specifically, to enhance the differentiation of uncertainty induced by AI-generatation, we fine-tune the model using a dataset comprising both natural and generated images. This approach aims to amplify the sensitivity of generated images to model perturbations while preserving the robustness of natural images against such perturbations. Consequently, this methodology widens the uncertainty gap between the model's responses to natural and generated images, thereby improving the distinguishability of their respective outputs.
>
>
>
> > Q2. The method's success appears to be reliant on the specific properties of DINOv2. The paper would be more impactful if it provided a deeper analysis into which properties of DINOv2 are crucial for this task to establish a more general principle.
>
> We appreciate your suggestion. We attribute DINOv2’s superior performance in distinguishing natural images from generated images, in part, to its self-supervised learning conducted exclusively on images, enabling robust capture of visual features. In our paper, we also evaluated WePe using CLIP as the feature extractor, where it exhibited reduced performance compared to DINOv2. This may result from CLIP’s reliance on text supervision during training, which results in image features that are predominantly semantic. In contrast, the distinction between natural and generated images hinges primarily on low-level features no semantic information, which DINOv2 captures more effectively owing to its image-only self-supervised training paradigm.
>
> > Q3. How does WePe differentiate between epistemic uncertainty arising from AI generation versus uncertainty from other natural distributional shifts (e.g., domain changes, resolution variations, or color-space differences)? Could you discuss potential strategies to mitigate confusion with non-generative OOD samples?
>
> Thanks for your constructive comments. We believe the following explanations would significantly improve the quality of our work.
> To alleviate the mentioned challenge, we first select DINOv2 as the feature extractor, as its training on a diverse set of natural images maximizes the inclusivity of the natural distribution, thereby reducing the likelihood of misclassifying natural distribution shifts. Second, when training is feasible, we further enhance the detection of AI-induced distribution shifts through distribution calibration, optimizing the method’s ability to distinguish generated images from natural ones.
>
> > Q4. If an attacker has white‐box access, they could conceivably craft images whose feature representations remain insensitive to weight perturbations, defeating the core detection signal. What defenses might mitigate such targeted attacks?
>
> We appreciate your inspiring comments. We believe this would be a promising direction in our community.
>
> In this regard, input randomization [1] serves as a potential defense against white-box adversarial attacks. During inference, applying random transformations to the input image, such as varying the image resolution, can disrupt the specific structure of adversarial perturbations, thereby enhancing the robustness of our method.
>
> Meanwhile, we find that advanced diffusion models provide a promising solution to mitigate adversarial attacks. However, this in turn leads to a further question: once a natural image is reconstructed by a diffusion model, should the reconstructed image be considered real or generated in our detection problem?
>
> We are looking forward to your further inspiring feedback in this promising direction.
>
> Reference:
>
> [1]: Mitigating adversarial effects through randomization. Xie et al. ICLR 2018

---

> ### Comment · Reviewer_frxx · 2025-08-05
>
> Thank you for the detailed rebuttal and the additional experiments in response to other reviewers, which have addressed my concerns—I will be maintaining my score.

---

> > ### Author Response · Authors · 2025-08-06
> >
> > We sincerely thank the reviewer for the valuable comments, which have significantly improved the quality of our paper. We also appreciate the reviewer’s acknowledgment of our efforts during the discussion. Should there be any further questions, we would be glad to address them to continue improving our work. Thank you for your time and thoughtful review.

---

### Comment · Area_Chair_XDz6 · 2025-08-03
**Author-Reviewer Discussion**

Dear Reviewers,

The author-reviewer discussion period is now open and will continue until August 6. Please review the authors’ rebuttal to determine whether it adequately addresses your concerns.  If you have further questions or comments, engage with the authors by acknowledging that you’ve read their response and providing additional feedback as needed

Sincerely,

Your AC

---

### Note · Authors · 2025-08-12

We sincerely thank all reviewers for their insightful feedback and valuable suggestions on our work. The reviewers consistently acknowledged the strengths of our method, including the novel framework that leverages epistemic uncertainty for AI-generated image detection, the training-free nature of WePe and its strong generalization to unseen generators, the method's theoretical foundation, the comprehensive experimental validation across multiple benchmarks, and the paper's clear presentation. We greatly appreciate the reviewers' recognition of these contributions, which further affirm the significance and practical value of our approach in addressing security challenges posed by generative models.

In response to the reviewers' suggestions, we have carefully addressed each point and made revisions to enhance the paper's quality. To tackle baseline comparisons, we added new experiments using advanced multimodal models and uncertainty baselines, demonstrating WePe's superior performance. We also analyzed multiple random perturbations and input perturbations for further gains, and visualized failure cases to illustrate feature space changes. Additionally, we expanded the discussion on uncertainty estimation in the related work section, incorporating key references, and elaborated on aleatoric and domain uncertainty, including estimation challenges. For improved clarity, we reorganized key tables from the appendix into the main text, added error bars, clarified data sources, corrected all typos and notation inconsistencies, and updated section titles. We also provided a tighter bound analysis for Eq. (13). These revisions reflect our deep respect for the reviewers' insights and our proactive commitment to rigorous improvements.

We are truly grateful for the reviewers' acknowledgment of our rebuttal, which addressed their concerns and led to score increases. We commit to fully incorporating these revisions into the final manuscript to further elevate its quality. Thank you again for your pivotal role in improving this work.

---

### Decision · Program_Chairs · 2025-09-17

**Decision:**

Accept (poster)

**Comment:**

This paper presents an uncertainty-based method for detecting AI-generated images. The underlying assumption is that there is a distribution shift between natural images and AI-generated images, such that models trained on natural images exhibit high epistemic uncertainty (EU) when processing AI-generated images. Building on this idea, the authors introduce a weight perturbation method applied to pre-trained DINOv2 models to measure EU. Extensive experiments demonstrate competitive performance across four benchmarks.

The paper received four reviews: one accept and three borderline accepts. Overall, reviewers acknowledge the authors’ insight into the distribution shift between natural and generated images, their effective use of this property for detection, and the extensive results demonstrating the method’s strong performance. However, reviewers also raise concerns, including limited technical novelty, the method’s potential failure on real images that differ from the training distribution, its heavy reliance on DINOv2 model properties, and the lack of comparison with other uncertainty quantification (UQ) methods.  Additional questions were raised about the sensitivity of the method to changes in perturbation distributions and to the data used to fine-tune the model. The authors’ rebuttal is extensive and detailed, with new results that largely address most of the reviewers’ concerns.